# FilMaster: Bridging Cinematic Principles and Generative AI for Automated Film Generation

**Kaiyi Huang**[1*]    **Yukun Huang**[1]    **Xintao Wang**[2†]    **Zinan Lin**[3]    **Xuefei Ning**[4]

**Pengfei Wan**[2]    **Di Zhang**[2]    **Yu Wang**[4]    **Xihui Liu**[1†]

[1]The University of Hong Kong    [2]Kling Team, Kuaishou Technology
[3] Microsoft Research    [4]Tsinghua University

## Abstract

Existing AI-based film generation systems can generate high-quality videos, but struggle to design expressive camera language and establish cinematic rhythm. This deficiency leads to templated visuals and unengaging narratives. To address these limitations, we introduce FilMaster, an end-to-end automated film generation system that integrates real-world cinematic principles to generate professional-grade, editable films. Inspired by professional filmmaking, FilMaster is built on two key **cinematic principles**: (1) camera language design by learning cinematography from extensive real-world film references, and (2) cinematic rhythm by emulating professional post-production workflows. For camera language, our **Multi-shot Synergized Camera Language Design** module introduces a novel **scene-level** Retrieval-Augmented Generation (RAG) framework. Unlike shot-level RAG which retrieves references independently and often leads to visual incoherence, our approach treats an entire scene, comprising multiple shots with a shared spatio-temporal context and narrative objective, as a single, unified query. This holistic query retrieves a consistent set of semantically similar shots with cinematic techniques from a large corpus of 440,000 real film clips. These references then guide an LLM to synergistically plan coherent and expressive camera language for all shots within that scene. To achieve cinematic rhythm, our **Audience-Aware Cinematic Rhythm Control** module emulates professional post-production, featuring a *Rough Cut* assembly followed by a *Fine Cut* process that uses simulated audience feedback to optimize the integration of video and sound for cinematic rhythm. Extensive experiments show superior performance in camera language and cinematic rhythm, paving the way for generative AI in professional filmmaking. Here is the Project Page

## 1 Introduction

Rapid advancements in generative AI (Kong et al., 2024; Yang et al., 2024) and (Multimodal) Large Language Models ((M)LLMs) (OpenAI, 2024; Team et al., 2023) have opened up new possibilities for automated film production (Zhang et al., 2025b). However, current automated film generation systems often produce content that is visually templated and rhythmically flat, falling short of professional cinematic standards. For instance, in terms of cinematography, methods like Anim-Director (Li et al., 2024), MovieAgent (Wu et al., 2025), LTX-Studio (LTX Studio, 2024) often generate incoherent or templated shots within a scene (*e.g.*, "static shot" in Figure 1), as their planning relies on LLM imagination without professional filmmaking knowledge. Furthermore, regarding cinematic rhythm, simply concatenating video clips with limited or desynchronized audio to create a final film (Wu et al., 2025; LTX Studio, 2024) often results in a flat and unengaging experience for the audience (Table 1). These limitations reveal a fundamental gap: current automated AI systems struggle to **understand and implement the core cinematic principles** (Bordwell et al., 2004) that are crucial for crafting effective narratives (Rabiger, 2013).

---

*Work done during an internship at Kling Team, Kuaishou Tech. † Corresponding author.

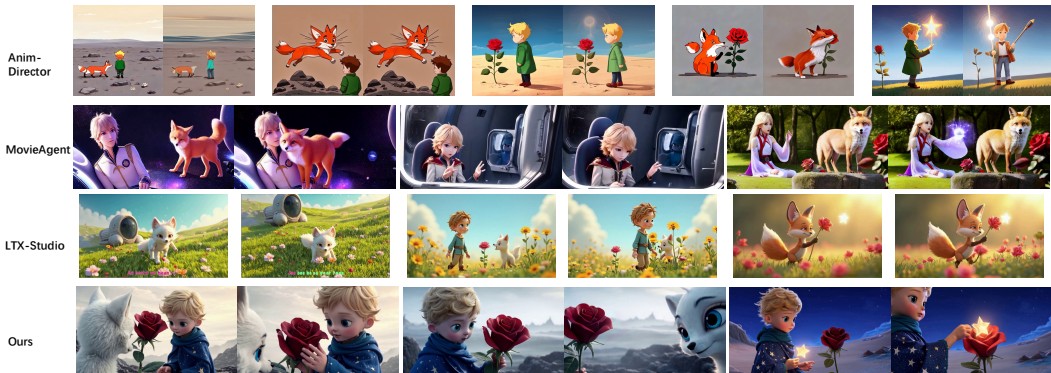

Figure 1: Qualitative comparison of film generation in the case of "little prince". Ours show more expressive camera language, compared to Anim-Director, MovieAgent, and LTX-Studio. More details are in Section 4.3.

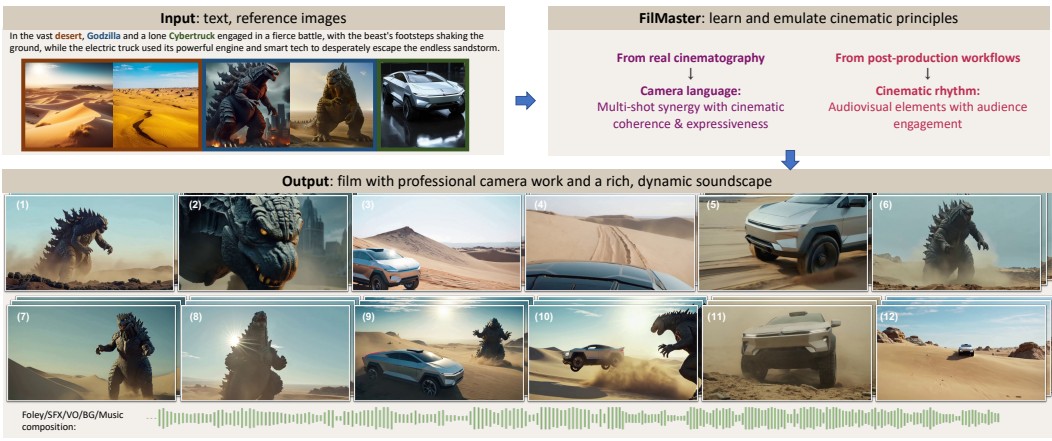

Figure 2: Video samples generated by FilMaster. Using the input of a text and reference images for characters and locations, FilMaster crafts high-quality films complete with professional camera language and cinematic rhythm, including rich, multi-layered audiovisual outputs (foley, sound effects (SFX), voice-over(VO), background ambiance, musical scoring, and video).

To address these challenges, we propose FilMaster, an end-to-end automated film generation system designed to bridge the gap from script to screen, as shown in Figure 2. Inspired by professional filmmaking, our approach is built upon two key **cinematic principles** (Bordwell et al., 2004): (1) Camera language design by learning cinematography from extensive real-world film references. Filmmakers traditionally hone camera language skills to improve narrative coherence and expressiveness by studying extensive film references. Drawing inspiration from this practice, FilMaster learns from a vast corpus of real film clips to apply professional camera language. (2) Cinematic rhythm control by emulating professional post-production workflows. FilMaster emulates the post-production workflow to control narrative pacing, and construct immersive audiovisual experiences, which are crucial for an engaging experience (Case, 2013). Guided by these principles, FilMaster's two core modules (Multi-shot Synergized Camera Language Design and Audience-Aware Cinematic Rhythm Control) drive a two-stage workflow for generation and post-production, yielding a fully editable and structured film (Figure 3).

Specifically, the innovation for camera language is our **Multi-shot Synergized Camera Language Design** module, which generates coherent and expressive cinematography. Our approach is centered on a novel **scene-level** Retrieval-Augmented Generation (RAG) framework that moves beyond the limitations of simpler, shot-level RAG methods. A shot-level RAG approach treats each shot as an independent query, retrieving cinematic references in isolation. This often leads to visual incoherence, where camera movements and styles change erratically between consecutive shots within the same scene (Figure 5). In contrast, our scene-level framework treats an entire scene, comprising multiple shots that share the same spatio-temporal context and narrative objective, as a single, unified query. This scene-level query is used to perform one retrieval operation from our database of 440,000 annotated film textual descriptions. The retrieved cinematic references then guide an LLM

Table 1: Comparison of film generation capabilities across different AI methods: Anim-Director Li et al. (2024), MovieAgent Wu et al. (2025), and LTX-Studio (commercial product) LTX Studio (2024).

| Method | Script Design | Camera Language Design | #Audio Types | A/V Sync | Video Adj. (Structure) | Video Adj. (Duration) | Audience Review | Editable Output |
|---|---|---|---|---|---|---|---|---|
| Anim-Director | ✓ | ✗ | 0 | ✗ | ✗ | ✗ | ✗ | ✗ |
| MovieAgent | ✓ | Templated | 1 | ✗ | ✗ | ✗ | ✗ | ✗ |
| LTX-Studio | ✓ | Templated | 1 | Limited | ✗ | ✗ | ✗ | ✓ |
| **FilMaster (Ours)** | ✓ | ✓(Film-referenced) | 5 | ✓ | ✓ | ✓ | ✓ | ✓ |

to synergistically re-plan the camera language for all shots within the scene. By enforcing a unified creative direction at the scene level, our method produces camera work that is both cinematically coherent and narratively expressive.

To achieve cinematic rhythm, we propose **Audience-Aware Cinematic Rhythm Control** module emulates a professional, audience-aware post-production workflow. This process is driven by MLLMs prompted to adopt distinct professional roles (*i.e.*, audience, film editor, sound designer): A *Rough Cut* is assembled by the generated video clips and then reviewed by an MLLM acting as a simulated audience. The MLLM can be conditioned on a specific demographic profile (*e.g.*, "short-drama audience") to generate targeted feedback. This feedback then guides the *Fine Cut* process, where MLLMs perform both video editing and sound design. Video editing involves structural and durational adjustments to video clips. Sound design ensures that a rich, multi-track soundscape is crafted, integrating diverse audio elements (background ambiance, musical scoring, voice-overs (VO), foley, and sound effects (SFX)) with multi-scale audiovisual synchronization. Through this Audience-Aware process, FilMaster adjusts narrative pacing and enables effective integration of audiovisual elements, resulting in outputs with immersive and engaging rhythms.

To summarize, our contributions are as follows:

- We propose FilMaster, an end-to-end automated film generation system that operationalizes **cinematic principles** for high-level creative control. It transforms the user input into a complete film, delivered as an editable and structured format compatible with professional production workflows.

- We introduce a novel **Scene-Level, Multi-shot Synergized Camera Language Design** for camera planning, that overcomes the limitations of static, templated or incoherent cinematography. By leveraging the Retrieval-Augmented Generation (RAG) approach with scene-level query on a large corpus of real films, it produces coherent and expressive camera language for each scene.

- We propose a **Audience-Aware Cinematic Rhythm Control** for cinematic rhythm. This mechanism simulates audience feedbacks to guide automated video editing and multi-scale sound design, resolving issues of flat pacing and disjointed audiovisuals.

- Experiments demonstrate that, compared to existing open-source and commercial film generation systems, our method achieves an improvement of 74.17% in camera language and 79.26% in cinematic rhythm, based on human ratings.

## 2 RELATED WORK

### 2.1 VIDEO GENERATION

Existing video generation models can be roughly categorized into diffusion model-based (Ho et al., 2022; Singer et al., 2022; Zhou et al., 2022; Khachatryan et al., 2023; Luo et al., 2023; Blattmann et al., 2023; He et al., 2022; Wang et al., 2023; Yang et al., 2024), and language model-based (Villegas et al., 2022; Chang et al., 2023; Kondratyuk et al., 2023; Yu et al., 2023a;b; Chang et al., 2022). Video diffusion models excel by progressively refining noisy inputs into clean video samples, with recent advancements like Sora (Brooks et al., 2024), HunyuanVideo (Kong et al., 2024), and Wan-Video (Wang et al., 2025) demonstrating remarkably high-quality visual synthesis via sophisticated latent diffusion techniques. Video language models, such as VideoPoet (Kondratyuk et al., 2023), are typically derived from the family of transformer-based language models that can flexibly incorporate multiple tasks in pretraining, and show zero-shot capabilities. Recent works (Guo et al., 2025;

Xiao et al., 2025; Cai et al., 2025), such as Veo3 (Google DeepMind, 2025), RunwayML (Runway AI, 2025), and MovieGen (Polyak et al., 2024) enable high-quality video generation. However, they require significant manual intervention for prompting or post-production for film with audiovisual elements. In contrast, our work provides a fully automatic, script-to-screen film generation system.

## 2.2 VIRTUAL CINEMATOGRAPHY

Virtual cinematography aims to incorporate cinematographic principles into both virtual and live-action videos (Christianson et al., 1996; Kardan & Casanova, 2008). The Virtual Cinematographer (VC) (He et al., 2023) introduces a paradigm for automatically generating complete camera specifications in real time for capturing events in 3D virtual environments. Other works focus on planning suitable camera placements for interactive tasks (Gleicher & Witkin, 1992; Mackinlay et al., 1990; Phillips et al., 1992). For example, Drucker et al. (Drucker et al., 1992; Drucker & Zeltzer, 1995; Drucker, 1994) formulate camera control as a constrained optimization problem to determine optimal camera positions for different shot types. While prior approaches primarily emphasize explicit camera placement strategies, our method instead learns and applies cinematographic principles from film data and post-production workflows, leveraging recent advances in video generation models and multimodal large language models (MLLMs).

## 2.3 LLMS FOR FILM PRODUCTION

Recent works for preliminary exploration in film production have begun to leverage the emerging reasoning and planning capabilities of LLMs (Wei et al., 2022; Lin et al., 2024; Zhang et al., 2025a). Anim-Director (Li et al., 2024) automates animation generation by employing large multimodal models (LMMs) to expand user inputs into coherent storylines, generate scene images and videos by iteratively optimization. The development of multi-agent systems (Park et al., 2023; Sun et al., 2024; Wu et al., 2023; Hong et al., 2024; Yuan et al., 2024; Huang et al., 2024) has further spurred innovation. For example, FilmAgent (Xu et al., 2025) proposes multi-agent collaboration through iterative feedback and revisions in constructed 3D virtual spaces. MovieAgent (Wu et al., 2025) multi-agent Chain of Thought planning for movie generation, which automatically structures scenes, camera settings, and cinematography. Distinct from these prior efforts, FilMaster significantly expands the role of LLMs to orchestrate a more comprehensive end-to-end film production pipeline, from pre-production through post-production. Crucially, our system uniquely integrates the learning of camera language from real film footage and the emulation of professional post-production workflows for cinematic rhythm control, aspects not holistically addressed by previous LLM-based film generation systems (Table 1).

## 3 METHOD

We introduce the overview of our system in Section 3.1, and detail the two core modules, Multi-shot Synergized Camera Language Design module in Section 3.2 and Audience-Aware Cinematic Rhythm Control module in Section 3.3, respectively. The pipeline is shown in Figure 3.

## 3.1 OVERVIEW OF FILMASTER

FilMaster is an end-to-end automated system for generating films. It takes a text and sets of reference images for characters and locations as input, and produces a complete film, delivered as a fully editable, multi-track timeline in the industry-standard format OpenTimelineIO (OTIO) (OpenTimelineIO, 2023). As illustrated in Figure 3, our framework operates in a two-stage process: Generation Stage and Coordination Stage.

Let $T$ denote the input text, $I_c$ the input set of character reference images, and $I_l$ the input set of location reference images. The overall process is a composition of two stages, Generation Stage ($\mathcal{G}$) and Coordination Stage ($\mathcal{C}$). First, raw video clips are generated:

$$V_{\text{clips}} = \mathcal{G}(T, I_c, I_l)$$

These clips are then processed in post-production into the final film, $\mathcal{F}_{\text{film}}$:

$$\mathcal{F}_{\text{film}} = \mathcal{C}(V_{\text{clips}})$$

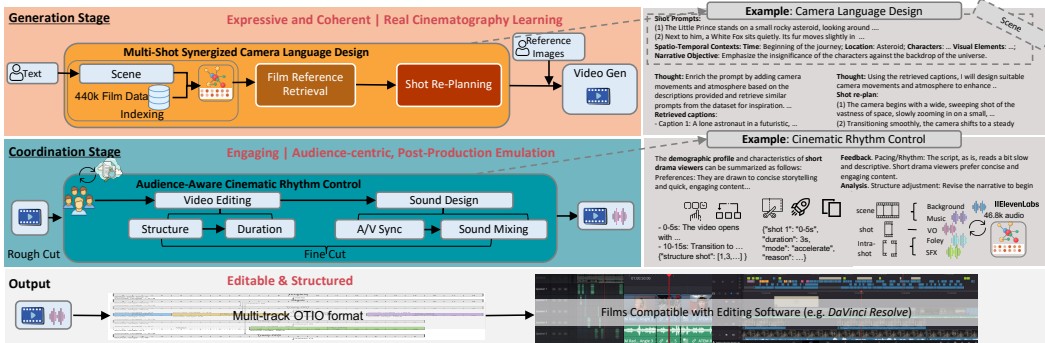

Figure 3: Overview of the FilMaster framework for end-to-end automated film generation. It processes user input to produce editable, structured outputs, guided by cinematic principles. FilMaster include the (1) Multi-shot Synergized Camera Language Design module within the Generation Stage, leveraging real film data for coherent and expressive visuals, and the (2) Audience-Aware Cinematic Rhythm Control module within the Coordination Stage, emulating professional post-production for enhanced audience engagement. Examples are shown in gray area.

where $\mathcal{F}_{\text{film}}$ is the final film represented as an OTIO timeline, comprising both the video track and the audio tracks.

**Generation Stage: From Input to Raw Footage.** This stage translates the inputs $(T, I_c, I_l)$ into a collection of raw video clips, $V_{\text{clips}} = \{V_1, \ldots, V_n\}$. It includes:

- *Script Elaboration*: LLMs first perform a coarse-to-fine refinement on the input text $T$, expanding it into $n$ detailed scenes with rich spatio-temporal contexts (detailed in Section A).
- *Camera Language Design:* The Multi-shot Synergized Camera Language Design module (see Section 3.2) then plans the camera language for each scene, referencing a large corpus of real film clips to ensure visual coherence and narrative expressiveness.
- *Video Generation:* A video generation model synthesizes the clips based on the planned camera language and the reference images $I_c$ and $I_l$ (see Section B).

**Coordination Stage: From Raw Footage to Final Film.** This stage assembles the generated clips $V_{\text{clips}}$ into a polished film $\mathcal{F}_{\text{film}}$ by video editing and sound design based on audience-aware feedback (see Section 3.3). This stage is driven by MLLMs prompted to adopt distinct professional roles.

- *Audience-Aware Review:* An MLLM, simulating a target audience, reviews an initial *Rough Cut* and generates structured feedback on its pacing, structure, and audio.
- *Video Editing:* Guided by this feedback, an LLM acting as an editor refines the visual sequence into a *Fine Cut* by adjusting shot order and duration.
- *Sound Design:* A multi-track soundscape (background ambiance, musical scoring, foley, voice-over, SFX) is composed by generation or retrieval and synchronized with the video to create a cohesive audio-visual experience.

The final film is a multi-track OTIO file, packaging both video and audio and ensuring seamless integration with professional post-production workflows.

## 3.2 Multi-shot Synergized Camera Language Design

To enable coherent and expressive camera language in the Generation Stage, we propose the Multi-shot Synergized Camera Language Design module that introduces a novel **scene-level** Retrieval-Augmented Generation (RAG) (Gao et al., 2023) framework. Different from shot-level RAG which causes incoherence by retrieving references independently for each shot (Figure 5), we propose a scene-level querying mechanism: multiple shots within the scene, which share the same spatio-temporal contexts and a unified narrative objective, are treated as a single query. This holistic query retrieves semantically similar shots from a large-scale dataset of 440,000 annotated textual descriptions of real film clips, then the retrieved shots with cinematic techniques are used to guide an LLM to re-plan camera language for each shot within a scene.

A **scene** $S$, from Script Elaboration, is the primary unit of planning, comprising shot prompts $P$, its spatio-temporal context $C$ and narrative objective $O$ (example shown in the upper right of Figure 3). Our reference dataset is a collection of $N$ film clips' annotated textual descriptions, $D_{\text{film}} = \{d_1, d_2, \ldots, d_N\}$. The entire process can be formalized as follows:

**Spatio-Temporal-Aware Indexing.** This indexing prepares a query vector of the scene and a vector database of film dataset. The scene $S$ is encoded into a query vector $\boldsymbol{q}$ using a text embedding model $E(\cdot)$:

$$\boldsymbol{q} = E(S) = E((P, C, O))$$

Concurrently, the entire film dataset of textual descriptions $D_{\text{film}}$ is split into small chunks, encoded into vectors $\{\boldsymbol{v}_i\}_{i=1}^N$, and stored in a vector database $V_{\text{db}}$ by applying the same embedding model to each textual description:

$$V_{\text{db}} = \{\boldsymbol{v}_i \mid \boldsymbol{v}_i = E(d_i), d_i \in D_{\text{film}}\}$$

**Film Reference Retrieval.** Given the query vector $\boldsymbol{q}$, we retrieve the top-$K$ most relevant cinematic examples from $V_{\text{db}}$ by finding the nearest neighbors in the embedding space. This is achieved by identifying the indices $\mathcal{I}$ of the $K$ vectors in $V_{\text{db}}$ based on semantic similarity to $\boldsymbol{q}$. The retrieved references $R$ are the set of annotated textual descriptions corresponding to these indices:

$$R = \{d_i \mid i \in \mathcal{I}\}$$

**Shot Re-planning.** The LLM re-plans ($\pi_{\text{re-plan}}$) detailed shot prompts $P$ for the entire scene with camera language based on film references, conditioning on both the original scene description $S$ and the retrieved references $R$. This is formulated as:

$$P_{\text{camera}} = \pi_{\text{re-plan}}(S, R)$$

The output $P_{\text{camera}}$ is a coherent sequence of shot descriptions for the scene $S$, specifying shot types, camera movements, angles, and atmospheric characteristics. We conduct the above for $n$ scenes separately. This scene-level shot design ensures visual continuity and narrative impact, a key distinction from shot-level generation approaches.

### 3.3 AUDIENCE-AWARE CINEMATIC RHYTHM CONTROL

In the Coordination Stage, the Audience-Aware Cinematic Rhythm Control module orchestrates the film's post-production by emulating a professional workflow with a sequence of specialized (M)LLMs. This process transforms the set of raw clips $V_{\text{clips}}$ into a final film $\mathcal{F}_{\text{film}}$.

**Audience-Aware Review.** This process generates actionable feedback by simulating a target audience's response. Let $A_{\text{target}}$ be the specified audience archetype (*e.g.*, "short-drama audience"). First, a *Rough Cut* $T_{\text{rough}}$ is assembled by concatenating the video clips $V_{\text{clips}}$ with placeholder audio descriptions. Next, an LLM leverages search tools to generate a detailed audience profile $P_{\text{audience}}$ based on the given archetype. An MLLM, acting as a simulated audience, then critiques the $T_{\text{rough}}$ conditioned on $P_{\text{audience}}$, producing a set of critiques $F_{\text{critique}}$. Finally, these critiques are translated into a structured set of actionable recommendations $R_{\text{actions}}$ by another LLM, covering structural organization, timing and duration, and audio coherence.

**Video Editing.** Based on the audience-aware review $R_{\text{actions}}$, this process refines the visual narrative to achieve a picture-locked *Fine Cut*, denoted as $V_{\text{fine}}$. Let $V_{\text{rough}}$ be the video track of $T_{\text{rough}}$. Guided by $R_{\text{actions}}$, the system applies a set of available editing operations $\mathcal{O} = \{\text{rearrange}, \text{trim}, \text{accelerate}\}$ to the rough video track. The output $V_{\text{fine}}$ is a finalized visual sequence with optimized narrative pacing and structural flow.

**Sound Design.** The sound design process constructs a rich, multi-track soundscape synchronized with the picture-locked video $V_{\text{fine}}$. It employs a multi-scale synchronization strategy. Audio tracks are then generated at their corresponding temporal scales based on $R_{\text{actions}}$: background ambiance and musical scoring for each scene, voice-overs for each shot, and foley and sound effects for each event. These individually generated tracks are then aggregated into a complete set of tracks, $A_{\text{tracks}}$, for the entire film. Finally, an automated mixing function produces the final audio track, $A_{\text{final}}$. The full implementation details of each composition step are provided in Section C and Section H. The final output $\mathcal{F}_{\text{film}}$ is composed of $V_{\text{fine}}$ and $A_{\text{final}}$ in OTIO format.

Table 2: Automatic evaluation of film generation on FilmEval of Camera Language (CL) and Cinematic Rhythm (CRh). * denotes a commercial product. † denotes that the same video model was applied.

| Metric | Anim-Director | Anim-Director† | MovieAgent | MovieAgent† | LTX-Studio* | Ours† |
|---|---|---|---|---|---|---|
| CL ↑ | 2.96 | 3.02 | 2.74 | 2.55 | 3.74 | **4.50** |
| CRh ↑ | 1.94 | 2.38 | 1.74 | 1.98 | 3.62 | **4.32** |

## 4 EXPERIMENTS

### 4.1 EXPERIMENT SETTING

**Implementation Details.** We use *GPT-4o* (OpenAI, 2024) for script generation, RAG, video editing, and sound design (VO, background, musical scoring). For audience-aware review and sound design (foley and sound effect), we employ *Gemini-2.0-Flash* (Team et al., 2023). We use Kling Elements 1.6 (Kling, 2025) as the video generation model, capable of incorporating multiple reference images as conditions. The generated video clips are at 1920×1080 resolution, comprising 153 frames per sequence. We use $K = 3$ for retrieval, text-embed-3-small (OpenAI, 2025) for embedding model.

**Evaluation Metrics.** To evaluate our method, we employ both automatic evaluation metrics and user studies. We follow MovieAgent (Wu et al., 2025) to adopt VBench (Huang et al., 2023b) as evaluation metrics (see Section F.3). However, VBench focuses on the generated quality, lacking the ability to evaluate camera language and cinematic rhythm of the generated films. Given the absence of existing metrics tailored for this task, we establish *FilmEval*, a holistic evaluation benchmark for both automatic evaluation and user studies. FilmEval is based on 6 high-level dimensions essential for assessing film quality: *Narrative and Script (NS)*, *Audiovisuals and Techniques (AT)*, *Aesthetics and Expression (AE)*, *Rhythm and Flow (RF)*, *Emotional and Engagement (EE)*, and *Overall Experience (OE)*. These dimensions are further decomposed into 12 specific dimensions (*e.g.*, Script Faithfulness, Narrative Coherence), and we use the combination of the twelve criteria to calculate Camera Language (CL) and Cinematic Rhythm (CRh) (detailed in Section E). For automatically evaluating the aspects, we use *Gemini-1.5-Flash* (Team et al., 2023) as the evaluation model designed to assess generated films across the defined dimensions. We validate the effectiveness of automatic evaluation by measuring its correlation with human judgments in Section F.4.

**Test Dataset.** Our evaluation employs a diverse set of 20 test cases comprising two distinct types: 10 cases from MoviePrompts (Wu et al., 2025), which feature detailed descriptions, with an average of 100.4 words; and 10 shorter, more concise prompts with an average of 15.2 words, specifically designed by annotators to evaluate our method's flexibility in handling varied input complexities.

**Comparing Models.** We compare our method against three automatic film generation systems: the animation generation method Anim-Director (Li et al., 2024), the movie generation agent MovieAgent (Wu et al., 2025), and the commercial platform LTX-Studio (LTX Studio, 2024). To ensure a fair comparison, we include their video generation models with Kling 1.6, the same video model used in our method for the open-source baselines. For the commercial platform LTX-Studio, while we could not change its internal video model, we included its default automated sound effects (SFX) in our evaluation to assess its end-to-end performance.

### 4.2 QUANTITATIVE RESULTS

**Automatic Evaluation.** As shown in Table 2, our method achieves an average improvement of 67.46%, with significant gains in camera language (49.92%) and cinematic rhythm (85.15%). This represents a substantial lead over existing methods like Anim-Director (+87.35%) and MovieAgent (+105.99%), which have deficiencies in camera planning and audiovisual coordination. We also outperform the commercial system LTX-Studio (+19.81%), which struggles with narrative pacing and coherence. When using the same base video model (denoted by †), our approach excels. Baselines still underperform our method by 65.15% and 94.49%, respectively. This is because: (i) The evaluation focuses on the overall quality including 12 dimensions instead of only visual quality; (ii) The baselines focus on the character consistency, but fail to effectively integrate the cinematic principles, which hinders the overall narrative coherence. Detailed results are available in Section F.1.

**User Study.** We conduct a user study to evaluate the quality of generated films. Five participants were instructed to read the criteria defined in FilmEval, and were asked to rate each video indepen-

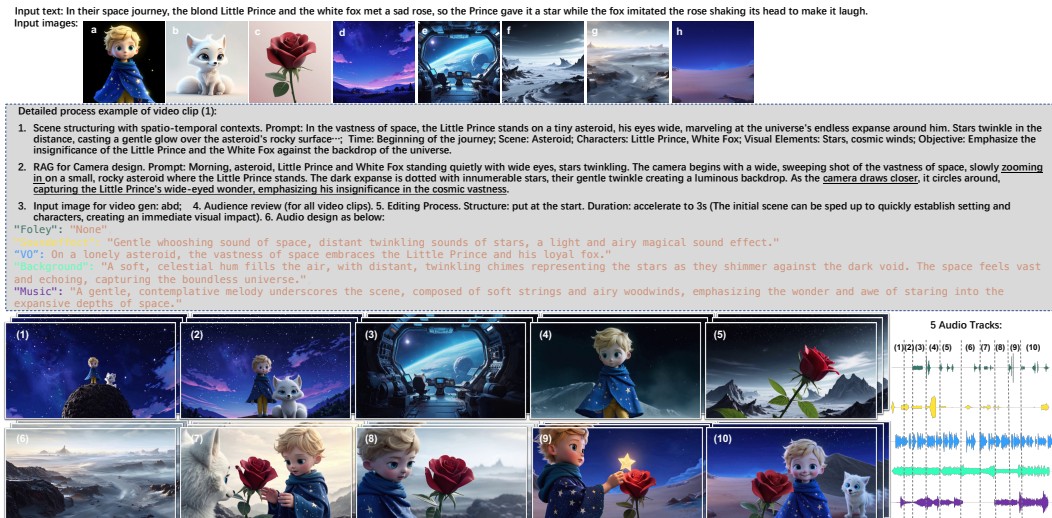

Figure 4: The input and corresponding video frame extractions from our FilMaster, illustrated alongside associated multi-track audio. The detailed process of the video clip (1) is illustrated in gray area.

Table 3: User study on film generation methods. * denotes commercial product. † denotes the same video model (Kling 1.6) applied.

| Method | NS ↑ | AT ↑ | AE ↑ | RF ↑ | EE ↑ | OE ↑ | CL ↑ | CRh ↑ | Avg ↑ |
|---|---|---|---|---|---|---|---|---|---|
| Anim-Director | 1.94 | 2.16 | 1.94 | 2.12 | 2.12 | 2.36 | 2.15 | 2.04 | 2.11 |
| Anim-Director† | 1.94 | 2.35 | 1.44 | 1.94 | 1.84 | 2.20 | 2.16 | 1.85 | 1.95 |
| MovieAgent | 1.57 | 1.63 | 1.70 | 1.70 | 2.20 | 2.27 | 1.66 | 1.83 | 1.84 |
| MovieAgent† | 1.32 | 2.38 | 1.68 | 2.02 | 1.96 | 1.92 | 2.01 | 1.89 | 1.88 |
| LTX-Studio* | 2.28 | 3.04 | 3.22 | 2.90 | 3.16 | 2.96 | 2.80 | 3.05 | 2.92 |
| **Ours†** | **3.70** | **3.80** | **3.80** | **3.73** | **3.93** | **3.87** | **3.76** | **3.82** | **3.79** |

dently. We randomly selected five cases from our dataset, compared our FilMaster with the other five methods (including †). In total, we collected 1,800 ratings, with 150 votes per evaluation criteria. We show the six dimensions in Table 3, and detailed results in Table 8. The results show that our FilMaster demonstrates superior performance in film generation compared to the baselines, with an average increase of 74.17% in camera language, and 79.26% in cinematic rhythm.

### 4.3 QUALITATIVE RESULTS AND COMPARISON

**Example.** As shown in Figure 4, our method generates descriptions with camera language and designs a multi-track audio based on the text prompt derived from the input text, forming a cohesive audiovisual narrative with camera language design and cinematic rhythm control. Additional examples are provided in Section F.6.

**Comparison.** (1) Camera Language: The baselines struggle to produce expressive cinematography, as shown in Figure 1. Anim-Director and MovieAgent generate static shots with fixed camera angles, failing to convey the narrative's progression. LTX-Studio, while visually appealing, also lacks deliberately designed camera movements to support the story's objectives. In contrast, our method utilizes a dynamic and purposeful camera language. As shown in row 4 of Figure 1, the camera actively participates in narrative expressiveness: In the first two images, the camera zooms in to a close-up of the Little Prince, the White Fox, and the Rose to show the interaction. In the middle two images, it then shifts focus between character pairs to sequentially highlight their actions of delighting the Rose. In the last two images, it performs another close-up to emphasize the actions of the Little Prince giving a star to the Rose. (2) Cinematic Rhythm: Limitations in rhythm and audio design exist across the baselines. Anim-Director lacks audio entirely, severely restricting its storytelling capabilities. MovieAgent only incorporates basic voice-overs, resulting in a flat auditory experience. LTX-Studio employs a black-box automated audio design with only one audio type,

**Objective**: Emphasize the **insignificance** of the Little Prince and the White Fox against the backdrop of the universe.

Prompt Summary: Morning, asteroid, Little Prince and White Fox standing quietly with wide eyes, stars twinkling.

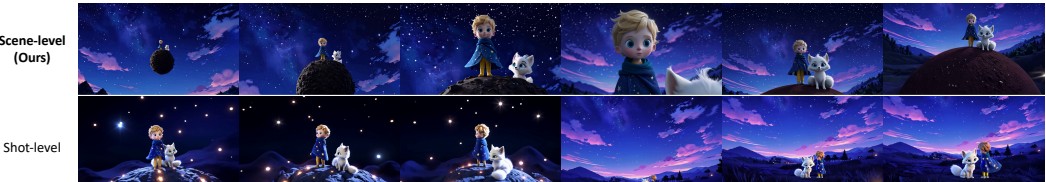

Figure 5: Ablation study of scene-level and shot-level RAG-based camera planning. **Our scene-level approach** (top) generates coherent and synergized sequences using coordinated movements like "zoom in", "close-up", and "pull back". **The shot-level approach** (bottom) produces disjoint sequences with erratic movements ("orbit", "pan across"), failing to maintain visual continuity.

Table 4: Ablation study on the effects of Camera Language Design Module and Cinematic Rhythm Control Module on overall performance (average across 12 criteria).

| Method | w/o Camera + Rhythm | w/o Rhythm | Ours |
|---|---|---|---|
| Avg ↑ | 3.75 | 4.17 | **4.67** |

which often leads to desynchronization between visual events and sound effects, and its narrative pacing can be slow and repetitive. In contrast, our method leverages audience-aware feedback to guide both video editing and sound design. By utilizing a multi-scale synchronization strategy, it creates a rich, coherent soundscape that is coupled with the visual pacing, resulting in a more immersive and engaging cinematic rhythm.

## 4.4 ABLATION STUDY

We ablate our method without Multi-shot Synergized Camera Language Design module and Audience-Aware Cinematic Rhythm Control in Table 4 on the case of "Little Prince" with 9 scenes and 16 shots. Quantitative results show that removing cinematic rhythm module significantly decreases the average score in FilmEval, highlighting its role in cinematic expression with similar generated content. We also examined the impact of Multi-shot Synergized Camera Language Design module, which helps to form a coherent generated content. We show the qualitative comparison of scene-level and shot-level RAG-based camera planning in Figure 5. Our scene-level generates coherent and synergized shots using coordinated movements like "zoom in", "close-up", and "pull back". The shot-level approach produces disjoint sequences with erratic movements ("orbit", "pan across"), failing to maintain visual continuity. We show detailed ablation study of our method in Section F.5, including the effectiveness of scene-level shot re-planning and RAG, the impact of audience-aware components, the influence of dataset size, the robustness of RAG on different LLMs.

## 5 CONCLUSION

We introduced FilMaster, an end-to-end automated film generation system designed for professional-grade film generation. FilMaster uniquely integrates cinematic principles, focusing on camera language design and cinematic rhythm control, while ensuring industry-compatible, editable outputs. We propose a Multi-shot Synergized Camera Language Design module that learns cinematography directly from a vast corpus of 440,000 real film clips. By designing a scene-level query mechanism and leveraging Retrieval-Augmented Generation (RAG), this module produces expressive, coherent camera plans with narrative objective. Our Audience-Aware Cinematic Rhythm Control module emulates professional post-production. This includes *Rough Cut* assembly, a *Fine Cut* process refined by simulated audience feedback, including video editing and sound design, all orchestrated to achieve compelling narrative flow and profound emotional impact. Furthermore, we proposed FilmEval, a comprehensive benchmark for assessing AI-generated films across six key cinematic dimensions. Extensive experiments demonstrate FilMaster's state-of-the-art performance, with an average improvement of 74.17% in camera language and 79.26% in cinematic rhythm based on human ratings, showcasing significant advancements in generating films with expressive visual language and engaging rhythm.

ETHICS STATEMENT

Our work utilizes an internal dataset of short clips derived from commercially available films for academic research, in accordance with fair use principles. To respect copyright, this dataset will not be publicly released; however, the pre-processing of film data will be open-source.

We acknowledge two primary ethical risks: 1) The model may inherit societal biases present in the source films. 2) As a generative tool, it has potential for misuse, such as fake news.

REPRODUCIBILITY STATEMENT

To ensure reproducibility, we provide the detailed descriptions of methodology in Section 3. For (M)LLM instructions used in our method, we provide in Section H. For the dataset used, we provide a description of its construction, content, and statistics in Section G.

ACKNOWLEDGEMENTS

This work is supported in part by the NSFC-RGC Joint Research Scheme through the Research Grant Council of Hong Kong under grant N_HKU76925, and in part by National Nature Science Foundation of China (No. 62561160156).

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

## A  SCRIPT ELABORATION

Given an input text, the script is hierarchically expanded. This coarse-to-fine approach ensures coherent progression from abstract concepts to concrete visual descriptions that can be translated into audiovisual content. The script elaboration unfolds in four progressive steps, each building upon the output of the previous step, implemented through a chain of specialized LLMs working sequentially. The LLM instructions and examples are shown in Section H.

- **Synopsis:** The input text is expanded into a synopsis that introduces theme, characters, and plots, forming the foundation for subsequent development. This step establishes the core narrative framework without detailed visual considerations.
- **Simplified Storyboard:** The synopsis is enriched with background information, causal chains, narrative developments, and outcomes, ensuring the story has appropriate twists while maintaining narrative coherence and logical consistency. This step transforms narrative concepts into structured story events.
- **Detailed Storyboard:** The simplified version is further elaborated into a detailed storyboard containing concrete visual narratives. This includes both primary shots (featuring main character actions with specific descriptions) and contextual shots (showing surrounding environments and establishing shots), alongside metaphorical imagery to enhance emotional resonance and thematic depth. This step translates story events into visual sequences.
- **Scene:** The detailed storyboard is segmented into distinct scenes that function as spatio-temporal contexts within one scene (*i.e.*, events occurring in the same location during the same timeframe), based on chronological order and spatial continuity. These scenes serve as fundamental units for subsequent audiovisual generation, ensuring consistent visual and narrative treatment across related shots. For each scene, we extract key elements including prompt, time, location, characters, visual elements, and narrative objectives (justifying each scene's narrative necessity for plot advancement or character development). These extracted elements collectively provide structured input for the Scene-level Multi-shot Camera Design module. Here detailed audio descriptions (VO, background ambiance, musical scoring) for each scene are generated as a placeholder for audience-aware review (VO) and scene-level sound design (background ambiance, musical scoring).

## B  VIDEO GENERATION

In the Generation Stage, the video clips ($V_{\text{clips}}$) are synthesized by a video generation model, which is conditioned on three inputs: the textual prompts from the shot re-planning process, a set of character reference images ($I_c$), and a set of location reference images ($I_l$). To ensure the correct visual references are used for each scene, an MLLM selects the appropriate $I_c$ and $I_l$ from a user-provided set based on the scene's character and location descriptions (described in Section A). For flexibility, if reference images are not provided by the user, they can be synthesized on-the-fly by an image generation model. All shots within the same scene are conditioned on the same reference images to maintain visual consistency.

## C  SOUND DESIGN

Directly relying on MLLMs to design multi-track audio (including background ambience, musical scoring, voice-over (VO), foley, and sound effects) often results in poor synchronization with video content. This is primarily due to the MLLM's limited capability in managing coordination across multiple audio layers and temporal alignment. To address this, we propose a multi-scale audio-visual synchronization strategy, which systematically designs different audio types at appropriate temporal resolutions. This process is further supported by retrieval-augmented generation (RAG) from curated audio libraries or synthesized voice content, followed by systematic audio preprocessing.

Specifically, synchronization is designed across three temporal scales:

- Scene-level synchronization: Background ambience and musical scoring are assigned at the scene level to establish atmosphere and emotional tone. This builds upon the rough audio design from Generation Stage.

- Shot-level synchronization: VO, including narration and dialogue, is designed at the shot level to maintain close alignment with visual content and enhance narrative clarity.
- Intra-shot synchronization: Foley and sound effects are assigned at a fine-grained temporal resolution. MLLM analyzes video content with second-level timecodes to detect specific events like actions, movements, and environmental elements that require sonic representation, ensuring detailed audiovisual coordination.

To ensure audio quality and contextual relevance, we employ the RAG approach for non-voice audio tracks, similar to Multi-shot Synergized Camera Language Design. We construct a curated audio library comprising 46,826 sound assets, including 5,877 music tracks and 40,949 other audio assets (atmospheric sounds, foley, and effects) sourced from the Internet. Each asset is tagged with semantic descriptors, emotional qualities, and acoustic properties to facilitate context-aware retrieval. For voice-over content, we utilize ElevenLabs' text-to-speech services (ElevenLabs, 2025), allowing for flexible and high-fidelity voice generation tailored to specific narrative demands.

Composing multiple audio tracks directly often leads to inconsistencies in loudness, frequency balance, and dynamic range. To mitigate these issues, we apply audio preprocessing techniques that ensure cross-track and track-video harmonization. This includes:

- Loudness units relative to full scale (LUFS) normalization to maintain consistent perceived loudness across tracks (*e.g.*, -16 LUFS for voice content, -28 LUFS for background elements).
- Frequency adjustment to avoid spectral clashes between different types of audio, particularly ensuring voice intelligibility by attenuating competing frequencies in background tracks.
- Dynamic equalization to enhance clarity and cohesion.

## D  ROBUSTNESS ANALYSIS OF FILMASTER

In this section, we evaluate the robustness of our system. Specifically, we use three mechanism to boost the robustness of our system: 1) separating review of *Rough Cut* and video editing; 2) adding audience perspective to help edit the videos; 3) disentangle multimodal information for (M)LLMs for processing information. We evaluate by ablating these components qualitatively and quantitatively. We randomly choose 10 cases, including 160 shots as test samples, and set breakpoints at the steps of structural organization, and timing and duration. We evaluate through two metrics: the corrected ratio and the success ratio. The corrected ratio is defined as whether they correct by comparing with the original ones. The success ratio is defined as whether the correction is correct (there is no contradictory in responses, no factural fact). We also show qualitative comparison in Figure 6 and Figure 7.

**Separating review of *Rough Cut* and video editing.**

We observed that if we combine review and correction together into a single MLLM, it will be overwhelmed with the task. The reasoning and analysis ability are therefore be impacted. Thus to keep the analysis, correction ability, we separate the review of *Rough Cut* and video editing. We found that by separating the clear task, the system becomes more robust and effective at correcting.

Table 5: Ablation study on the robustness of our system. We evaluate the Corrected Ratio and Success Ratio across 10 test cases.

| Method | Corrected Ratio (%) | Success Ratio (%) |
|---|---|---|
| Analysis and Correct | 60 | 50 |
| w/o Audience Perspective | 50 | 50 |
| **Ours (Full System)** | **100** | **100** |

**Adding audience perspective to help edit the videos.**

We observed that if we remove the perspective of audience, the post-production stage will be dominated by the perspective of director in the generation stage: There is no correction of the *Rough*

*Cut*, as it will be considered good enough in the view of director. This director-only perspective hampers the effective correction, and cannot be adjusted to meet the requirements of different types of audiences.

**Disentangling multimodal information for (M)LLMs for processing information.**

Asking a single MLLM to simultaneously process and reason about diverse multimodal inputs (video, audio, text) is a significant challenge, even for state-of-the-art models like Gemini. This complexity often leads to hallucinations or the neglect of specific requirements. To mitigate this, we break down the task by disentangling multimodal information. Instead of feeding all inputs into one model, our approach first uses an MLLM to generate detailed textual captions of the video. Then, a separate LLM uses these captions, alongside the script and placeholder audio descriptions, to perform the complex reasoning and editing (detailed in Figure 21-Figure 23). This separation of visual perception (captioning) from cognitive reasoning (editing) dramatically reduces complexity. As shown in Table 6, while a direct approach attempts all corrections, it fails to do successfully (0% success ratio). Our disentangled method, however, maintains a high success ratio, proving its effectiveness for complex post-production tasks.

Table 6: Ablation study on disentangling multimodal information. We evaluate the effect on the Corrected and Success Ratios across 10 test cases.

| Method | Corrected Ratio (%) | Success Ratio (%) |
|---|---|---|
| Directly addressing multimodal info | 100 | 0 |
| **Ours (Separating multimodal info)** | **100** | **90** |

## E    FILMEVAL EVALUATION INSTRUCTIONS

FilmEval is based on six high-level dimensions essential for assessing film quality: *Narrative and Script (NS)*, *Audiovisuals and Techniques (AT)*, *Aesthetics and Expression (AE)*, *Rhythm and Flow (RF)*, *Emotional and Engagement (EE)*, and *Overall Experience (OE)*. These dimensions are further decomposed into twelve specific criteria for detailed evaluation:

- **NS**: Script Faithfulness (SF), Narrative Coherence (NC)

- **AT**: Visual Quality (VQ), Character Consistency (CC), Physical Law Compliance (PLC), Voice/Audio Quality (V/AQ)

- **AE**: Cinematic Techniques (CT), Audio-Visual Richness (AVR)

- **RF**: Narrative Pacing (NP), Video-Audio Coordination (VAC)

- **EE**: Compelling Degree (CD)

- **OE**: Overall Quality (OQ)

While our work highlights two key modules for camera language and cinematic rhythm, it's important to recognize that cinematic quality arises from the holistic integration of various elements. Our evaluation dimensions are therefore designed to capture not only the direct outputs of each module but also their synergistic impact on the final film:

- The Multi-shot Synergized Camera Language Design is primarily evaluated through NS (SF, NC), ensuring visual storytelling aligns with the script, and through key visual aspects of AT (VQ, CC, PLC), reflecting the quality and coherence of the planned visual foundation. This module also lays the groundwork for effective AE (CT) by designing shots with inherent cinematic qualities and contributes to the visual component of AE (AVR).

- The Audience-Aware Cinematic Rhythm Control module's effectiveness is measured by audio-related aspects of AT (V/AQ), the realized AE (CT, AVR) through sophisticated editing and sound design, the mastery of RF (NP, VAC), and the resulting EE (CD). This module coordinates the visual and auditory elements into a cohesive and impactful rhythmic experience, evaluated by the ultimate arbiter OE (OQ).

Derived metrics (CL, CRh) are computed from preceding base metrics: CL is calculated as the average of (SF, NC, VQ, CC, PLC) + 0.5 * average of (CT, AVR), CRh is calculated as the average of (V/AQ, NP, VAC, CD, OQ) + 0.5 * average of (CT, AVR).

To reduce subjectivity in evaluation, we set up the criteria with more observable metrics, such as "1 point: Chaotic and disorderly — exhibits $\geq 3$ major logical problems". We show the detailed criteria of FilmEval for both automatic evaluation and user study as follows:

---

**NARRATIVE AND SCRIPT**

**Script Faithfulness**
**1 point:** Severely deviates from the original script — $\geq 50\%$ of key scenes missing or replaced, two or more major character attributes altered, and three or more plot elements contradict the original.
**2 points:** Partially follows the original script — $< 50\%$ of key scenes preserved, 1–2 major inconsistencies in character settings, and at least two identifiable deviations from core plot events.
**3 points:** Generally follows the original script — $\geq 70\%$ of key scenes preserved, character settings largely consistent with minor trait changes, deviations limited to nonessential details that do not affect the main storyline.
**4 points:** Highly faithful to the original script — $\geq 90\%$ of key scenes accurately represented, all major character settings preserved, only minor omissions without plot contradictions.
**5 points:** Completely faithful to the original script — all key scenes, character settings, and relevant details correctly reproduced; no detectable deviations.

**Narrative Coherence**
**1 point:** Chaotic and disorderly — exhibits $\geq 3$ major logical problems (clear causal errors, contradictory temporal order, character actions contradict prior information, or plot discontinuities) that prevent coherent understanding.
**2 points:** Basically understandable but flawed — contains $\geq 2$ clear logical gaps or missing transitions that noticeably disrupt narrative logic.
**3 points:** Generally coherent — main storyline is clear; may contain one minor logical deficiency (e.g., underdeveloped motivation) that does not impede overall comprehension.
**4 points:** Smooth and coherent — flows naturally with reasonable plot development, clear cause-effect links, and only negligible logical issues.
**5 points:** Completely coherent — natural, well-justified plot development with no logical holes; all cause-effect relationships are clear.

**Reordered Structure:**
**1.Scene 1: Office Environment**
1. 00:00-00:05
2. 00:05-00:10

**2.Scene 2: Looking Out the Window**
1. 00:10-00:15
2. 01:15-01:20
3. 00:15-00:20
4. 00:20-00:25
5. 00:25-00:30
6. 00:30-00:35

**3.Scene 3: Restaurant Interaction**
1. 00:35-00:40
2. 00:40-00:45
3. 00:45-00:50
4. 00:50-00:55
5. 00:55-01:00

**4.Scene 4: Personal Reflection**
1. 01:00-01:05
2. 01:05-01:10
3. 01:10-01:15
4. 01:20-01:25
5. 01:25-01:26

(a) Combine review and correction together

Scene 1: Introduction and Initial Focus
**Scene 1: Office Introduction**
- 00:00-00:05: A fluorescent light flickers above a computer screen. A young woman with curly hair appears and begins to type. She wears a tank top and necklace.
- 00:05-00:10: Close-up shot of the woman at her desk, typing on the keyboard. Files and a desk lamp are seen on the desk in front of her.

**Scene 2: Captivated by the Window**
- 00:10-00:15: A shot of the woman in her office from behind her. The bright light coming through the window makes everything blurry and hard to see. She's looking out the window.
- 00:15-00:20: Close up of the city seen from the woman's office, with her looking over the buildings from a high point.
- 00:20-00:25: A view of the office from the corner, showing the woman sitting at her desk looking towards the window.
- 00:25-00:30: Close up of the woman's side profile, looking out the window with a necklace and small hoop earrings.
- ...

(b) Without audience perspective

Structure after reordering and analysis:
**1.Scene 1: The Routine Office Environment**
1. **00:00-00:05**: A fluorescent light flickers above a computer screen. Sarah, a young woman with curly hair, appears and begins to type. She wears a tank top and necklace.
2. **00:05-00:10**: Close-up shot of Sarah at her desk, typing on the keyboard. Files and a desk lamp are seen on the desk in front of her.

**2.Scene 2: Longing for the Outside World**
1. **00:25-00:30**: Close up of Sarah's side profile, looking out the window with a necklace and small hoop earrings.
2. **00:20-00:25**: A view of the office from the corner, showing Sarah sitting at her desk looking towards the window.
3. **00:10-00:15**: A shot of Sarah in her office from behind her, with bright light making everything blurry as she looks out the window.

**3.Scene 3: Daydreams of the City**
1. **00:15-00:20**: Close up of the city seen from Sarah's office, with her looking over the buildings from a high point.
2. **00:25-01:26**: The view from where Sarah was standing to the bright window with a sun flare.

**4.Scene 4: Cafe Interaction**
1. **00:40-00:45**: Medium shot of Sarah with curly hair and a man with gray hair and a beard as they sit in a restaurant near a window.
2. **00:45-00:50**: Medium shot of Sarah, the man, and a third person to the right side of the shot.
3. **00:50-00:55**: Close up of Sarah and the man at the restaurant table, with her speaking to the man.
4. **00:55-01:00**: Close up of Sarah and the man at the restaurant table, with her smiling and speaking to the man while gesturing with her hands.

**5.Scene 5: Emotional Growth and Reflection**
1. **00:30-00:35**: Close up of Sarah's side profile, looking out the window with a necklace and small hoop earrings.
2. **01:00-01:05**: Close up of Sarah's face and upper torso, as she flips through a drawing calendar.
3. **01:05-01:10**: Sarah turns and smiles at the camera. The next clip shows a woman carrying a camera and wearing a red skirt.
4. **01:20-01:25**: Sarah is standing and then turns to look at the camera with a smile.

(c) Ours (separate review and correction, w audience perspective)

Figure 6: **Qualitative comparison of "Separating review of Rough Cut and video editing", and "Adding audience perspective".** We illustrate the case of "Her gaze was fixed on that window" as an example for structural organization. For (a), combining review and correction together will lose the ability to make correction, directly copying the original order. For (b), without audience perspective, the system tend to keep the order designed by director in the generation stage, failing to make effective correction. For (c), ours show effective correction and logical explanation of the adjusted structure.

(a) Directly addressing multimodal info

```
[
{
"shot 1": "00:00-00:05",
"duration": "2s",
"mode": "crop",
"reason": "Start shorter to show office monotony
without dwelling. Focus on the light's harshness and
her early sense of confinement."
},
{
"shot 2": "00:05-00:10",
"duration": "3s",
"mode": "retain",
"reason": "The change in lighting will show different
feel in the atmosphere, the desk environment still
depicts monotonousness."
},
{
"shot 3": "00:10-00:15",
"duration": "2s",
"mode": "crop",
"reason": "Reduce time spent here; emphasizes
Sarah's longing but avoid lingering too long on office
setting. A quick glance is enough."
},
{
"shot 4": "00:15-00:20",
"duration": "2s",
"mode": "crop",
"reason": "Cut down on the static desk view to
maintain a faster tempo. Focus on the window view."
},
{
"shot 5": "00:20-00:25",
"duration": "3s",
"mode": "retain",
"reason": "Allow more time to show the city view,
emphasize how the bustling environment outside
contrasts with Sarah's routine."
},
```

(b) Ours (Separating multimodal info)

```
Begin the script generation: Her gaze was fixed on
that window.
Response of duration: [
{
"shot 1": "00:00-00:05",
"duration": 4,
"mode": "accelerate",
"reason": "The initial flickering light and pulling back
shot sets the scene, but accelerating slightly maintains
the sterile office feel without losing impact."
},
{
"shot 2": "00:05-00:10",
"duration": 3,
"mode": "crop",
"reason": "Close-up of the typing scene should be
tightened to avoid monotony and emphasize the
contrast of lighting and work clutter."
},
{
"shot 3": "00:10-00:15",
"duration": 5,
"mode": "retain",
"reason": "This scene introduces the personal touches
and longing, so it should remain at full length to
develop emotional depth."
},
{
"shot 4": "00:15-00:20",
"duration": 3,
"mode": "crop",
"reason": "The scene of office supplies and posters
can be shortened as it reiterates previous information
about the workspace."
},
{
"shot 5": "00:20-00:25",
"duration": 4,
"mode": "accelerate",
"reason": "Speeding up the view of the city outside
emphasizes her fixation on the world outside the
office without losing the shot's intent."
},
```

Figure 7: **Qualitative comparison of "Separating multimodal information".** We illustrate the case of "Her gaze was fixed on that window" as an example for timing and duration. For (a), directly addressing multimodal information will let the model hallucinate with response, and fails to give propor correction reason. For instance, it tries to choose the *retain* mode that keeps the duration as original 5s, but give the instruction of reducing duration to 3s. For (b), separating multimodal information, the model can give self-consistent response without contradictory.

## AUDIO-VISUALS AND TECHNIQUES

**Visual Quality**
**1 point:** Severely broken visuals — multiple critical failures ($\geq 3$) such as missing objects, heavy distortion, or broken frames; key elements unrecognizable.
**2 points:** Obvious visual flaws — $\geq 2$ scenes with missing or distorted elements; artifacts noticeably disrupt viewing.
**3 points:** Basically complete visuals — all essential elements present; occasional minor errors or brief artifacts that do not affect comprehension.
**4 points:** Clear and complete visuals — very few minor imperfections; no major missing or distorted elements.
**5 points:** Flawless visuals — all elements correctly rendered throughout; no observable distortions or artifacts.

**Character Consistency (visual)**
**1 point:** Severely inconsistent designs — $\geq 2$ major appearance attribute changes (e.g., face shape, hairstyle, clothing) across scenes; same character may appear as different individuals.
**2 points:** Noticeable fluctuations — character features change across multiple scenes; identity recognizable but inconsistently portrayed.
**3 points:** Generally consistent — appearance largely stable; occasional minor inconsistencies apparent only on close inspection.
**4 points:** Highly consistent — stable features across nearly all scenes and angles; any inconsistencies are negligible.
**5 points:** Perfectly consistent — precise character features maintained in all scenes and actions; no observable fluctuations.

**Physical Law Compliance**
**1 point:** Severely violates physics — $\geq 3$ extreme violations (impossible movements, wrong gravity, unrealistic collisions) that break realism.
**2 points:** Multiple violations — at least two clear physics errors causing noticeably unrealistic motion or effects.
**3 points:** Generally compliant — most motions follow physical expectations; mild artificiality in some movements but acceptable.
**4 points:** Good compliance — natural movements and believable interactions; only minimal deviations.
**5 points:** Perfect compliance — movements, collisions, and effects behave in line with real-world physics; no anomalies.

**Voice/Audio Quality**
**1 point:** Extremely poor audio — voiceovers unclear or missing; sound effects chaotic or severely distorted; audio impairs comprehension.
**2 points:** Poor audio — voiceovers occasionally unclear; sound effects minimal or poorly synchronized; noticeably below standard.
**3 point:** Average audio — voiceovers generally clear; sound effects appropriate but not refined.
**4 points:** Good audio — clear, well-mixed voiceovers and rich sound effects that effectively support scenes.
**5 points:** Excellent audio — highly clear, expressive voiceovers and nuanced, well-synchronized sound design with no noticeable flaws.

## AESTHETICS AND EXPRESSION

**Cinematic Techniques**
**1 point:** Single, stiff shots — static, repetitive framing with no purposeful use of basic film language (composition, shot scale, angle).
**2 points:** Limited shot variation — some shot types used but movements are stiff or poorly executed; film language inconsistent or ineffective.
**3 points:** Common techniques used adequately — standard shots (close, medium, wide) and generally smooth camera movement; film language correct but not distinctive.
**4 points:** Rich film language — good variety of shots used intentionally; smooth, natural camera movements that enhance narrative or emotion.
**5 points:** Highly creative and precise techniques — wide, inventive range of shots and precise camera execution; film language is expressive and purposeful.

**Audio-Visual Richness**
**1 point:** Extremely limited expression — monotonous, repetitive visual/audio elements with minimal variation or layering.
**2 points:** Basic and formulaic — attempts at expression exist but are simple and predictable, with little stylistic diversity.
**3 points:** Moderate diversity — some scenes show varied styles or tempos; overall richness uneven and lacking full coherence.
**4 points:** Visually and sonically expressive — multiple techniques used cohesively to create layered meaning, mood shifts, or stylistic nuance.
**5 points:** Exceptionally rich audio-visual language — diverse, inventive, and highly expressive use of sound and visuals that forms a distinctive artistic voice and strong narrative/emotional impact.

## RHYTHM AND FLOW

**Narrative Pacing**
**1 point:** Completely uncontrolled pacing — multiple extreme pacing issues ($\geq 3$: abrupt skips, overly long pauses, rushed key events) that severely impair comprehension.
**2 points:** Obviously inconsistent pacing — at least two clear pacing problems (e.g., rushed or overly elongated plot points) that disrupt rhythm.
**3 points:** Generally appropriate pacing — reasonable progression with minor inconsistencies (one or two scenes slightly off) that do not hinder understanding.
**4 points:** Well-controlled pacing — natural scene durations and transitions; good balance between tension and relief.
**5 points:** Precisely controlled pacing — deliberate timing that enhances emotional impact and narrative clarity; smooth transitions between tempo changes.

**Video-Audio Coordination**
**1 point:** Severely unsynchronized — persistent audio-video mismatches, repeated lip-sync errors of multiple frames, and frequent sound-action mismatches that disrupt viewing.
**2 points:** Obvious lack of synchronization — recurring lip-sync or timing mismatches and poor coordination between voice and visuals.
**3 points:** Basically synchronized — audio and video mostly aligned with occasional minor mismatches that do not interfere with viewing.
**4 points:** Good coordination — voice, effects, and visuals are well-aligned; synchronization errors are rare and minor.
**5 points:** Perfect synchronization — all sound elements precisely match visual actions and lip movements; creates a harmonious viewing experience.

## EMOTIONAL AND ENGAGEMENT

**Compelling Degree**
**1 point:** No appeal — viewers cannot become immersed or emotionally connected; content fails to engage.
**2 points:** Insufficient appeal — weak emotional rendering; difficult to maintain viewer attention.
**3 points:** Basic appeal — generates some interest but lacks depth for strong emotional resonance.
**4 points:** Strong appeal — effective emotional rendering that elicits clear emotional responses and sustained interest.
**5 points:** Extremely compelling — powerful emotional tension and engagement that fully immerses viewers and produces strong resonance.

## OVERALL EXPERIENCE

**Overall Quality**
**1 point:** Extremely poor overall quality — $\geq 3$ core dimensions severely deficient; major issues impair comprehension or viewing value.
**2 points:** Poor overall quality — at least two major dimensions below acceptable standard; viewing value limited.
**3 points:** Average overall quality — most dimensions moderate or acceptable; balanced strengths and weaknesses; basic viewing value.
**4 points:** Good overall quality — majority of dimensions well-executed and cohesive; high viewing value with only minor issues.
**5 points:** Excellent overall quality — all major dimensions perform at a high level; consistent, precise, and artistic coordination yielding very high viewing and artistic value.

Table 7: Automatic evaluation of film production generation across 12 evaluation criteria on FilmEval, and 2 derived evaluation criteria for camera language (CL) and cinematic rhythm (CRh). [*] denotes commercial product. [†] denotes the same video model applied. Blue represents the derived metric for CL, and green stands for CRh .

| Method | NS ↑ | | AT ↑ | | | | AE ↑ | | RF ↑ | | EE ↑ | OE ↑ | Derived ↑ | | Avg ↑ |
|---|---|---|---|---|---|---|---|---|---|---|---|---|---|---|---|
| | SF | NC | VQ | CC | PLC | V/AQ | CT | AVR | NP | VAC | CD | OQ | CL | CRh | |
| Anim-Director | 1.60 | 2.20 | 4.20 | 3.45 | 3.55 | 1.00 | 3.05 | 2.50 | 2.10 | 1.00 | 2.45 | 2.30 | 2.96 | 1.94 | 2.45 |
| Anim-Director[†] | 1.60 | 2.60 | 4.20 | 3.80 | 3.20 | 2.00 | 2.80 | 2.60 | 2.20 | 2.80 | 2.20 | 2.40 | 3.02 | 2.38 | 2.70 |
| MovieAgent | 1.50 | 1.60 | 4.10 | 3.40 | 3.40 | 1.00 | 2.70 | 2.20 | 1.60 | 1.00 | 2.20 | 2.20 | 2.74 | 1.74 | 2.24 |
| MovieAgent[†] | 1.40 | 1.60 | 4.00 | 3.20 | 2.80 | 1.80 | 2.40 | 2.20 | 1.40 | 2.40 | 2.00 | 2.00 | 2.55 | 1.98 | 2.27 |
| LTX-Studio[*] | 2.50 | 3.00 | 4.95 | 4.10 | 3.90 | 3.10 | **4.10** | 3.85 | 3.15 | 4.10 | 3.65 | 3.75 | 3.74 | 3.62 | 3.68 |
| **Ours**[†] | **3.90** | **4.60** | **5.00** | **5.00** | **4.40** | **3.80** | **4.10** | **4.10** | **4.40** | **5.00** | **4.20** | **4.40** | **4.50** | **4.32** | **4.41** |

# F EXPERIMENTAL RESULTS

## F.1 QUANTITATIVE RESULTS ON AUTOMATIC EVALUATION

We provide the detailed automatic evaluation on twelve dimensions in FilmEval based on Gemini-1.5-Flash (Team et al., 2023). The results are shown in Table 7, with an average improvement of 67.46% in our FilMaster: 49.92% in camera language and 85.15% in cinematic rhythm, respectively. Our analysis reveals that existing approaches like Anim-Director (Li et al., 2024) and MovieAgent (Wu et al., 2025) significantly underperform across multiple dimensions including NS, AE, RF, EE, and OE. These methods demonstrate particularly severe deficiencies in audio quality and video-audio coordination. In contrast, our proposed method achieves substantial improvements across all evaluation dimensions in FilmEval, with an average performance increases of 87.35% and 105.99% compared to Anim-Director and MovieAgent, respectively. When compared with the commercial product LTX-Studio, we observe that LTX-Studio struggles with script faithfulness, narrative coherence, narrative pacing, and audio quality, likely due to insufficient integration of camera language and audiovisual elements. Our approach outperforms LTX-Studio with an average improvement of 19.81%, demonstrating the effectiveness of our film generation system. As for the same video base model comparison (denoted as [†]), the results show that the baselines underperform our approach by 65.15% and 94.49%, because: (i) The evaluation focuses on the overall quality including 12 dimensions instead of only visual quality; (ii) The baselines focus on the character consistency, but fail to effectively integrate the cinematic principles, which hinders the overall narrative coherence. The results show that our method outperforms existing methods (including the ones with the same video model) in all aspects, emphasizing the effectiveness of integrating cinematic principles into the automatic film generation system.

## F.2 QUANTITATIVE RESULTS ON HUMAN EVALUATION

We show the detailed results of user study in Table 8. Our method shows superior performances compared with other existing film generation systems in human evaluation across 12 dimensions, observations similar to Section F.1.

## F.3 QUANTITATIVE RESULTS ON VBENCH

We follow (Wu et al., 2025) to report the results of VBench (Huang et al., 2023b) in Table 9. (1) The results demonstrate that our method achieves the highest average score, primarily driven by its exceptional performance in generating dynamic visuals. Our method significantly outperforms all baselines on the "Dynamic Degree" metric, achieving a score of 0.9412. This result quantitatively validates the effectiveness of our Multi-shot Synergized Camera Language Design module in generating diverse and expressive camera movements. (2) Regarding other metrics such as "Aesthetic Quality" and "Motion Smoothness", our method performs competitively with other systems that use the same video generation model (Kling 1.6). This is expected, as these low-level visual attributes are predominantly determined by the generation model, rather than the high-level planning modules that represent our primary contribution (camera language and cinematic rhythm). (3) It is crucial to note, however, that VBench is primarily designed to evaluate low-level video generation quality. It currently lacks the capability to assess high-level cinematic aspects such as the narrative coherence of camera language or the effectiveness of cinematic rhythm and editing. Therefore, they do

Table 8: Quantitative comparison of different methods across 12 evaluation criteria. The "Avg" column shows the average score for each method. * denotes commercial product. † denotes the same video model applied. Values in bold indicate the best performance in that column.

| Method | NS ↑ | | AT ↑ | | | | AE ↑ | | RF ↑ | | EE ↑ | OE ↑ | Avg ↑ |
|---|---|---|---|---|---|---|---|---|---|---|---|---|---|
| | SF | NC | VQ | CC | PLC | V/AQ | CT | AVR | NP | VAC | CD | OQ | |
| Anim-Director | 2.08 | 1.80 | 2.52 | 2.24 | 2.32 | 1.56 | 2.20 | 1.68 | 2.64 | 1.60 | 2.12 | 2.36 | 2.09 |
| Anim-Director† | 2.04 | 1.84 | 2.60 | 2.56 | 2.48 | 1.76 | 1.68 | 1.20 | 2.16 | 1.72 | 1.84 | 2.20 | 2.01 |
| MovieAgent | 1.73 | 1.40 | 1.67 | 1.73 | 1.73 | 1.40 | 1.93 | 1.47 | 2.00 | 1.40 | 2.20 | 2.27 | 1.74 |
| MovieAgent† | 1.48 | 1.16 | 2.76 | 2.64 | 2.36 | 1.76 | 1.76 | 1.60 | 2.04 | 2.00 | 1.96 | 1.92 | 1.95 |
| LTX-Studio* | 2.44 | 2.12 | 3.16 | 3.00 | 2.84 | 3.16 | 3.52 | 2.92 | 2.60 | 3.20 | 3.16 | 2.96 | 2.92 |
| **Ours†** | **3.73** | **3.67** | **3.87** | **3.93** | **3.53** | **3.87** | **3.73** | **3.87** | **3.93** | **3.53** | **3.93** | **3.87** | **3.79** |

Table 9: Quantitative comparison of different methods on VBench. The "Avg" column shows the average score for each method in terms of aesthetic quality, background consistency, dynamic degree, motion smoothness, subject consistency. * denotes commercial product. † denotes the same video model applied.

| Method | Aesthetic Q. ↑ | BG Consist. ↑ | Dynamic Deg. ↑ | Motion Smooth. ↑ | Subject Consist. ↑ | Avg ↑ |
|---|---|---|---|---|---|---|
| Anim-Director | **0.6980** | 0.8410 | 0.1500 | 0.9638 | 0.7239 | 0.6753 |
| Anim-Director† | 0.6808 | **0.9611** | 0.0866 | **0.9943** | **0.9515** | 0.7349 |
| MovieAgent | 0.5800 | 0.9390 | 0.3500 | 0.9882 | 0.7053 | 0.7125 |
| MovieAgent† | 0.6118 | 0.9503 | 0.2000 | 0.9942 | 0.7041 | 0.6921 |
| LTX-Studio* | 0.6303 | 0.9464 | 0.4591 | 0.9919 | 0.6471 | 0.7350 |
| **Ours** | 0.6352 | 0.9375 | **0.9412** | 0.9848 | 0.8939 | **0.8785** |

not fully capture the core strengths of our system in automated filmmaking. The advantages of our method are demonstrated in our qualitative comparisons, the proposed quantitative comparisons and user studies.

## F.4 HUMAN CORRELATION COMPARISON

To validate our proposed automatic evaluation metric, we assess its correlation with human judgments. Following established practices (Park et al., 2021; Huang et al., 2023a), we compute three correlation coefficients across twelve criteria in Table 10: Pearson's $r$, Kendall's $\tau$, and Spearman's $\rho$. The results show that the correlation of the proposed automatic evaluation achieves an average of 0.5837, demonstrating high alignment with human judgments. We compare our automatic evaluation against several baselines, as shown in Table 11. Our proposed evaluation metrics, Gemini for 12 dimensions (Gemini-12dim), consistently achieve the highest correlation across all three coefficients: Pearson's $r$ (0.6185), Spearman's $\rho$ (0.6013), and Kendall's $\tau$ (0.5312). It significantly outperforms both existing domain-specific metrics like VBench and other MLLMs like GPT-4o, which show a much weaker correlation. Furthermore, we provide an ablation study on the impact of the number of evaluation dimensions: as the number of dimensions increases from 1, 6, to our proposed 12, the correlation with human perception consistently improves. This indicates that the comprehensive, multi-faceted nature of our 12-dimension metric is crucial for capturing the nuances of human judgment, which simpler metrics overlook.

## F.5 ABLATION STUDY

To assess the contribution of each component of our framework, we conducted a user study with 15 participants. For each experiment, we generated videos using our full model and several ablated versions. Participants were asked to select up to two videos that they felt best adhered to a specific criterion (either "camera language" or "cinematic rhythm"). We report the preference rate (*i.e.*, the percentage of total votes received) for each model variation in Table 12.

**Effectiveness of Shot Re-planning, RAG and Scene-Level.** We evaluate the core components of our Generation Stage. Participants were asked to select videos with the most professional and effective camera language, given the narrative objective designed by scene. As shown in Table 12, the version without shot re-planning saw its preference rate drop by 47.75% compared to the full model. The impact of the RAG module was even more pronounced, with its removal causing an

Table 10: Quantitative results of correlation coefficients across 12 evaluation criteria in FilmEval and their average in terms of Pearson Correlation $r$, Spearman's $\rho$, and Kendall's $\tau$ Coefficient ($p$-value $< 0.01$). Green represents the average of each correlation coefficient across 12 evaluation criteria.

| Correlation | NS | | AT | | | | AE | | RF | | EE | OE | |
| | SF | NC | VQ | CC | PLC | V/AQ | CT | AVR | NP | VAC | CD | OQ | Avg |
|---|---|---|---|---|---|---|---|---|---|---|---|---|---|
| Pearson's $r$ (↑) | 0.6119 | 0.6576 | 0.4678 | 0.3891 | 0.5317 | 0.7333 | 0.6884 | 0.7470 | 0.6308 | 0.6918 | 0.6718 | 0.6007 | 0.6185 |
| Spearman's $\rho$ (↑) | 0.5589 | 0.6073 | 0.4602 | 0.3886 | 0.5328 | 0.7386 | 0.7203 | 0.7149 | 0.5867 | 0.6791 | 0.6469 | 0.5808 | 0.6013 |
| Kendall's $\tau$ (↑) | 0.4870 | 0.5164 | 0.4260 | 0.3444 | 0.4845 | 0.6584 | 0.6415 | 0.6384 | 0.5047 | 0.5946 | 0.5715 | 0.5067 | 0.5312 |

Table 11: Comparison of correlation coefficients between model predictions and human judgments. Higher values indicate stronger correlation and better performance. Values in bold indicate the best performance for each metric.

| Metric | GPT-4o | VBench (avg) | Gemini-1dim | Gemini-6dim | **Gemini-12dim (Ours)** |
|---|---|---|---|---|---|
| Pearson's $r$ (↑) | 0.2194 | 0.5452 | 0.4393 | 0.4724 | **0.6185** |
| Spearman's $\rho$ (↑) | 0.1385 | 0.4536 | 0.5485 | 0.4911 | **0.6013** |
| Kendall's $\tau$ (↑) | 0.1116 | 0.3574 | 0.3822 | 0.4067 | **0.5312** |

83.43% decrease in preference. This shows the effectiveness of referencing proper cinematography from a large corpus of film data. Shot re-planning at the shot-level decreases by 73.97% compared to scene-level, which shows the effectiveness for synergized shots within at the scene-level. These results underscore the critical role of RAG-based scene-level shot re-planning in generating high-quality camera work. Figure 8 (row 1-4) shows the qualitative comparison for the scene with the narrative objective, which requires emphasizing the insignificance of the Little Prince and the White Fox against the universe. W/o shot re-planning and w/o RAG utilize a more static camera language, w/ RAG at the shot-level produces more disjoint and incoherent shots ("orbit" then "pan across", lack the coherence between shots visually), while ours (w RAG at the scene-level) utilizes a richer and more coherent camera language to represent the scene: (1) "a wide sweeping shot of vastness of space", (2) "slowly zooming in on a small rocky asteroid where the Little Prince stands", (3) "camera shifts to a steady close-up of the White Fox", and (4) "camera pulls back showing the duo's smallness against the universe".

**Impact of Audience-Aware Components.** We analyze the effectiveness of our Coordination Stage designed to tailor the output for a specific audience. We use ablated versions by removing the audience profile, the video editing pipeline (structural reorganization and duration adjustment), and the sound design. When participants evaluated the videos for cinematic rhythm, our full method was consistently preferred. It outperformed the various ablated versions by a margin ranging from 34.21% to 83.42%, demonstrating that our Audience-Aware Cinematic Rhythm Control is essential for creating a compelling final product.

**Influence of Dataset Size.** We investigate the relationship between data scale in Generation Stage and perceived output quality. The model using our full dataset was strongly preferred over those using smaller subsets. It received 94.35% more preference votes than a model using only 0.1k samples and an average of 44.56% more votes than models using 1k-10k samples. This highlights the significant benefits of a large-scale, high-quality dataset. Figure 8 (row 1, 4-6) presents a qualitative comparison for a scene where the narrative objective is to convey the insignificance of the characters. With only 0.1k reference samples, the method fails to maintain scene coherence, incorrectly transitioning to a different scene. Increasing the dataset to 1k and 10k samples improves coherence, but the resulting camera language is suboptimal and fails to effectively portray the characters' insignificance.

**Robustness of RAG Method on Different LLMs.** To validate that the effectiveness of our RAG method is not dependent on a single LLM, we test it with two different LLM backbones: DeepSeek-V3 (Liu et al., 2024) and LLaMA-3 70B (Touvron et al., 2023). Across these different LLMs, our RAG-enhanced method maintains a strong performance, achieving an average preference improvement of 55% over a non-RAG baseline. This demonstrates the generalizability and robustness of our proposed method.

**Using Open-Source Models.** To validate our method's generalization to open-source models, we conduct two experiments: 1) replace Kling 1.6 video generation model with Wan 2.1-VACE (Jiang

Table 12: Ablation studies and component analysis. We evaluate the impact of key components by removing them one at a time ("w/o"). The best-performing setting in each group is highlighted in **bold**.

| Ablation | Score ↑ |
|---|---|
| *Ablation on Shot Re-plan, RAG, and Scene-Level* | |
| w/o Re-plan | 0.267 |
| w/o RAG | 0.100 |
| w/ RAG, Shot-Level | 0.133 |
| **w/ RAG, Scene-Level (Ours)** | **0.511** |
| *Ablation on Audience Dimensions* | |
| w/o Profile | 0.250 |
| w/o Structure | 0.063 |
| w/o Timing | 0.210 |
| w/o Audio | 0.100 |
| **Ours** | **0.380** |
| *Effect of Dataset Size* | |
| 0.1k Samples | 0.026 |
| 1k Samples | 0.300 |
| 10k Samples | 0.210 |
| **440k Samples (Ours)** | **0.460** |
| *Effect of Different LLMs for RAG* | |
| Baseline | 0.180 |
| DeepSeek-V3 | 0.360 |
| LLaMA-3 70B | **0.450** |

Table 13: Ablation study on other video generation model and MLLM model. Our method can generalize to other backbones. * denotes commercial product. Blue represents the derived metric for CL, and green stands for CRh .

| Method | NS ↑ | | AT ↑ | | | | AE ↑ | | RF ↑ | | EE ↑ | OE ↑ | Derived ↑ | | Avg ↑ |
|---|---|---|---|---|---|---|---|---|---|---|---|---|---|---|---|
| | SF | NC | VQ | CC | PLC | V/AQ | CT | AVR | NP | VAC | CD | OQ | CL | CRh | |
| MovieAgent | 1.60 | 2.20 | 4.20 | 3.45 | 3.55 | 1.00 | 3.05 | 2.50 | 2.10 | 1.00 | 2.45 | 2.30 | 2.96 | 1.94 | 2.45 |
| Anim-Director | 1.50 | 1.60 | 4.10 | 3.40 | 3.40 | 1.00 | 2.70 | 2.20 | 1.60 | 1.00 | 2.20 | 2.20 | 2.74 | 1.74 | 2.24 |
| LTX-Studio* | 2.50 | 3.00 | 4.95 | 4.10 | 3.90 | 3.10 | 4.10 | 3.85 | 3.15 | 4.10 | 3.65 | 3.75 | 3.74 | 3.62 | 3.68 |
| **Ours (Wan2.1-VACE)** | 4.00 | 4.33 | 4.33 | 4.67 | 4.00 | 4.33 | 3.33 | 3.33 | 4.00 | 4.33 | 3.67 | 4.00 | **4.11** | **3.94** | **4.03** |
| **Ours (Qwen-VL)** | 4.00 | 4.50 | 5.00 | 5.00 | 4.00 | 4.00 | 3.50 | 4.50 | 4.00 | 5.00 | 4.50 | 4.50 | **4.42** | **4.33** | **4.38** |

et al., 2025) model. 2) replace Gemini (MLLM) with Qwen-VL (Bai et al., 2025) model. We show the automatic evaluation on FilmEval in Table 13.

### F.6 QUALITATIVE RESULTS

**Results of Multi-shot Synergized Camera Language Design.** We show qualitative results of scene-level synergized shots for certain narrative objective, detailed with the re-planned shots in Figure 10 and Figure 11. In Figure 10, the scene is composed of four shots (from Script Elaboration) with the narrative objective of "introducing Godzilla's presence and power". The shots are re-planned synergistically with camera language: (1) a wide-angle shot to show the environment, the camera panning upward to emphasize Godzilla's towering height; (2) a close-up shot of its razor-sharp eyes, the camera zooming in to accentuate the depth and focus of its gaze; (3) the camera pulling back to capture the powerful echoing; (4) a tracking shot to mirror its pace, the handheld camera to show the raw power and weight of its stride. In Figure 11, the scene, composed of two shots, aims to show the narrative objective of "showcasing the Cybertruck's maneuvering to seek safety". The shots are re-planned with camera language by: (1) a lower angle to capture the struggle and resilience against sands; (2) the camera shifting to a bird's eye view to show its maneuvering into the narrow gap.

**Objective**: Emphasize the **insignificance** of the Little Prince and the White Fox against the backdrop of the universe.

Prompt Summary: Morning, asteroid, Little Prince and White Fox standing quietly with wide eyes, stars twinkling.

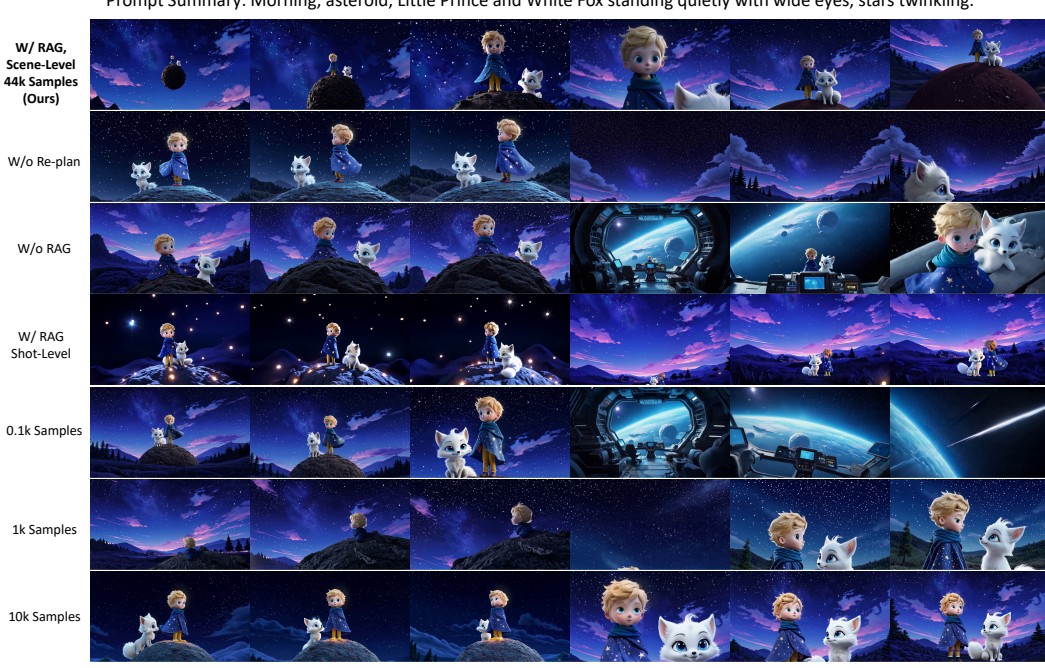

Figure 8: The qualitative results of the ablation study (effectiveness of shot re-planning and RAG, and influence of dataset size.) Ours (w RAG and 44k samples) shows the alignment with the narrative objective of the scene, which emphasizes the insignificance of the Little Prince and the White Fox against the backdrop of the universe. **For the effectiveness of shot re-planning and RAG (row 1-4), ours utilizes the richer and coherent camera language to represent the narrative objective**: (1) "begins with a wide, sweeping shot of the vastness of space", (2) "slowly zooming in on a small, rocky asteroid where the Little Prince stands", (3) "camera shifts to a steady close-up of the White Fox sitting next to him", and (4) "camera then pulls back, showcasing the duo's smallness against the infinite universe". **For the influence of dataset size (row 1, 4-6), ours conveys an optimal camera language while maintaining scene consistency**: with only 0.1k reference samples, the method fails to maintain scene coherence, incorrectly transitioning to a different scene. Increasing the dataset to 1k and 10k samples improves coherence, but the resulting camera language is suboptimal and fails to effectively portray the characters' insignificance.

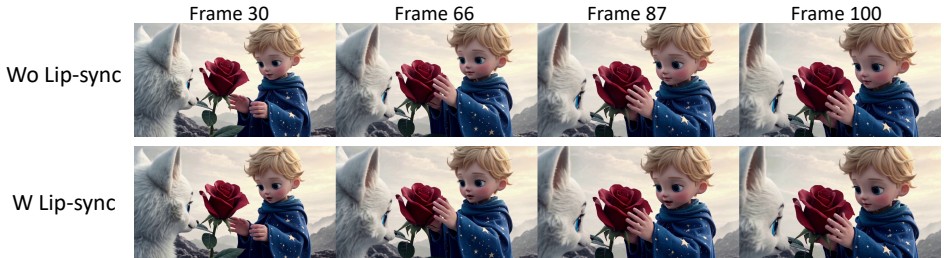

VO: Look, it's starting to shimmer, like us.

Figure 9: Qualitative results of the lip synchronization. Our method can be easily extended to incorporate an off-the-shelf synchronization module after the sound design to achieve lip synchronization.

**Comparison with Baselines.** We present a qualitative comparison with baseline methods in Figure 12. Compared to our method, the baselines tend to produce static camera language. To ensure a fair comparison, we equip the baselines with the same video generation model as ours (Kling 1.6). The results show that this primarily improves motion quality and fine-grained details (*e.g.*, hand generation), rather than introducing more dynamic cinematic techniques. This demonstrates that the cinematic techniques are improved by our proposed Multi-shot Synergized Camera Language Design, not just the underlying video model.

**Objective**: Introduce Godzilla's presence and power.

Prompt Summary: Day, desert, Godzilla looming and roaring, grounds shakes.

Shot 1: Begin with a **wide-angle shot** of the desert's edge, **capturing the shimmering heat rising off the sand**. Gradually, Godzilla's imposing silhouette appears on the horizon, with **the camera panning upward to emphasize its towering height**. Its rough, aged scales catch the sunlight, glinting like ancient armor and casting intimidating shadows across the landscape.

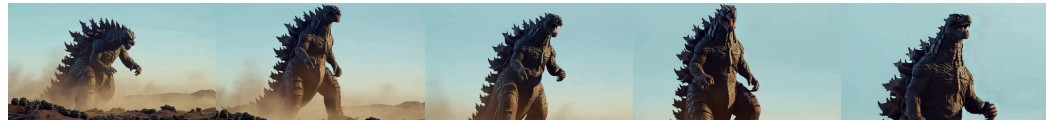

Shot 2: Transition to an intense **close-up shot of Godzilla's eyes, razor-sharp and full of intent**. The camera **zooms in dramatically**, accentuating the depth and focus of its gaze as it scans the barren landscape with a purpose that suggests deliberate evaluation.

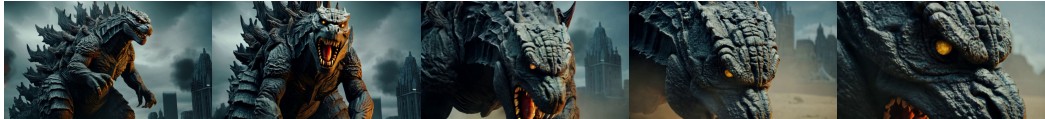

Shot 3: As Godzilla releases its thunderous roar, **the camera pulls back to capture the powerful reverberation echoing through the arid environment**. The roar's impact is heightened with a subtle slow motion effect, and the ground visibly trembles, sending ripples through the sand as distant dunes shake.

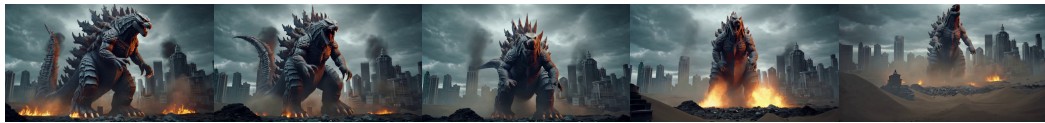

Shot 4: Follow Godzilla's steps with a **tracking shot that mirrors its deliberate pace**. The ground vibrates with each massive footfall, **and the handheld camera movement communicates the raw power and weight of its stride**. Dust swirls up around its feet, enhancing the tense atmosphere that accompanies Godzilla's advancing presence.

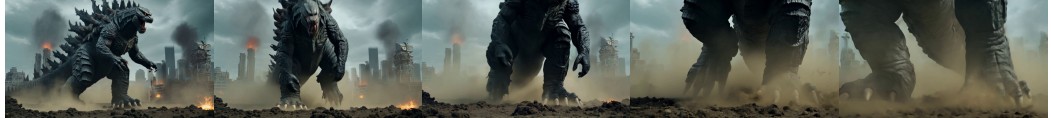

Figure 10: **Qualitative results of Multi-shot Synergized Camera Language Design**. The scene is composed of four shots (from Script Elaboration) with the narrative objective of "introducing Godzilla's presence and power". The shots are re-planned in a synergized manner with camera language.

**Results of the Final Generated Films.** We provide more generated films in Figure 13, Figure 14, Figure 15, and Figure 16. The qualitative results show that our proposed FilMaster can provide end-to-end film generation process, accomplishing a coherent film results based on the input of text, reference images of characters, and locations. Our proposed method can also handle different length of input texts and different number of reference images.

**Results of Lip-Synchronization.** Our method can be easily extended to incorporate an off-the-shelf synchronization module after the sound design to achieve lip synchronization. Here we use Kling model (Kling, 2025) to synchronize the lip of the Little Prince, with the VO "Look, it's starting to shimmer, like us", as shown in Figure 9.

## G  FILM DATASET

To ground our system's cinematographic knowledge in real-world examples, a large-scale film dataset was constructed for internal research purposes. The source films were segmented into short clips of 5 and 10 seconds in duration. Then an MLLM (LLaVA (Xu et al., 2024)) was employed to generate textual descriptions for each clip. These descriptions capture key cinematic elements, including camera language, character actions, and scene setting, forming the basis of our retrieval database. Our final dataset consists of approximately 310,000 5-second clips and 136,000 10-second clips. The generated textual descriptions have an average length of 231 words per clip.

**Objective**: Showcase the Cybertruck's maneuvering to seek safety.

Prompt Summary: Day, desert, Cybertruck evading, finds rock shelter, low visibility.

Shot 1: As the Cybertruck navigates through the fierce sandstorm, the **camera follows closely from a lower angle, capturing the struggle and resilience of the vehicle against the swirling sands**. The atmosphere around is tense and chaotic, with the sound of the wind howling and sand grains pelting against the truck's body, creating a sense of urgency and endurance. The visibility is low, with grains of sand creating a hazy curtain, but the Cybertruck's silhouette is clearly highlighted against the dusty, stormy backdrop.

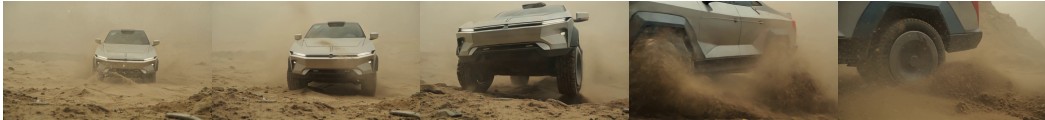

Shot 2: **The camera shifts to a bird's eye view as the Cybertruck discovers a narrow gap** between towering rock formations and carefully maneuvers into this protective enclave. The atmosphere shifts from chaotic to slightly calmer, with the howling wind subdued within the rock barriers, giving a sense of relief and safety. Dust particles still dance in the filtered light, creating a serene yet dramatic effect, as the Cybertruck comes to a rest in a safe position.

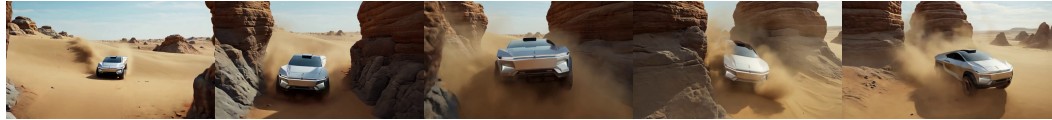

Figure 11: **Qualitative results of Multi-shot Synergized Camera Language Design.** The scene, composed of two shots, aims to show the narrative objective of "showcasing the Cybertruck's maneuvering to seek safety". The shots are re-planned with camera language synergistically.

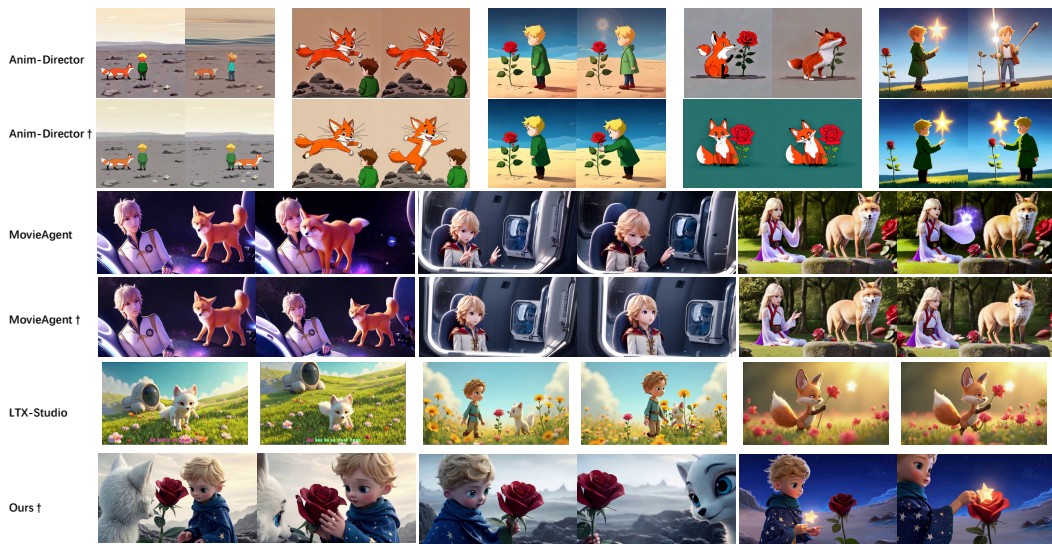

Figure 12: Qualitative comparison of film generation in the case of "little prince". Ours show more expressive camera language, compared to Anim-Director, MovieAgent, and LTX-Studio. † denotes the same video model (Kling 1.6) applied.

# H (M)LLM INSTRUCTIONS

We provide the prompts used in (M)LLMs for illustration. For script elaboration, we provide the LLM with the examples in Figure 17 and Figure 18. For Multi-shot Synergized Camera Language Design, we show in Figure 19. For Audience-Aware Cinematic Rhythm Control, we provide audience-aware review in Figure 20, video editing in Figure 21- Figure 23, sound design in Figure 24 and Figure 25.

# I DISCUSSION ON CINEMATIC STYLES

**Varied Cinematic Styles.** Our method can generate varied cinematic styles. Retrieval is used to enhance cinematic coherence, such as maintaining proper shot grammar, but does not override the style

**Input text:**

In their space journey, the blond Little Prince and the white fox met a sad rose, so the Prince gave it a star while the fox imitated the rose shaking its head to make it laugh.

**Input images:**

Asteroid    Spacecraft    Barren Planet    Little Prince    White Fox    Sad Rose

Figure 13: Qualitative results of the final generated film for the input text of "Little Prince".

**Input text:**

In the vast desert, Godzilla and a lone Cybertruck engaged in a fierce battle, with the beast's footsteps shaking the ground, while the electric truck used its powerful engine and smart tech to desperately escape the endless sandstorm.

**Input images:**

Desert    Godzilla    Cybertruck

Figure 14: Qualitative results of the final generated film for the input text of "Cybertruck battle with Godzilla".

instructions given to the model. The generator can balance retrieved cues with style-conditioning inputs, ensuring creative freedom. Increasing retrieval diversity (*e.g.*, larger K) further reduces potential bias. We demonstrate this flexibility through two examples in Figure 26. First, we show

**Input text:**

As Riley enters her teenage years, her mind's Headquarters undergoes a sudden transformation, welcoming new Emotions. Anxiety takes charge, believing that constant worry will help Riley navigate adolescence, while Envy fuels self-doubt. Joy, Sadness, Anger, Fear, and Disgust struggle to adapt as Anxiety's influence grows, sidelining their roles. As Riley faces mounting pressure, the imbalance leads to emotional turmoil. Realizing that Anxiety alone cannot define Riley's experience, the original Emotions work to restore balance. In the end, Riley embraces the complexities of growing up, with both old and new Emotions learning to coexist.

**Input images:**

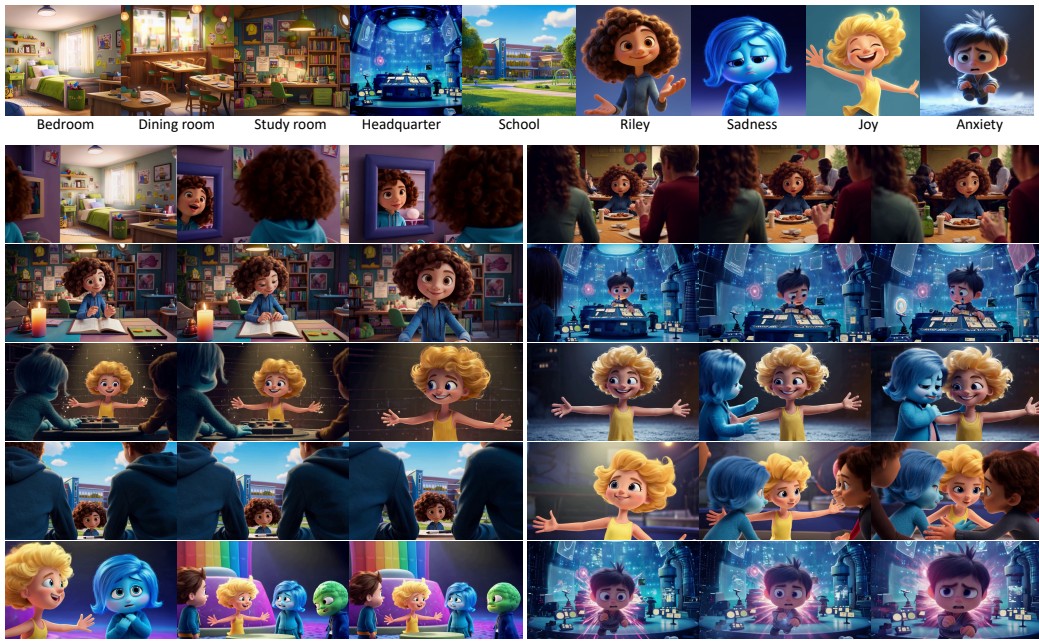

Bedroom   Dining room   Study room   Headquarter   School   Riley   Sadness   Joy   Anxiety

Figure 15: Qualitative results of the final generated film for the long input text of "Riley".

stylistic adaptation by transforming reference images into a "Ghibli style" using an off-the-shelf model, effectively shifting the outputs from generic animation to a specific Ghibli look. Second, we demonstrate that increasing the number of retrieved examples to K=10 allows for more relevant video clips, thereby diversifying the camera design. For instance, given the narrative objective "emphasize the insignificance of the Little Prince and the White Fox," the model yields a "close-up then pull back" with K=3, whereas it produces a "pan around with wide camera angle" with K=10.

**Generalization to Unseen Domains.** Our method demonstrates generalization capabilities in unseen domains, such as the documentary genre. Quantitative and qualitative results are presented in Table 14 and Figure 27, respectively. Given the input text "A documentary look at life in the desert and how living things survive the extreme environment," our method first elaborates a script featuring relevant characters serving as observers, such as a local biologist, a wildlife photographer, a storytelling Bedouin elder, and a passionate botanist. Subsequently, it generates videos incorporating appropriate camera language and cinematic rhythm.

Table 14: Quantitative comparisons on documentary *v.s.* non-documentary genres.

| Method | CL (↑) | CRh (↑) | Avg (↑) |
|---|---|---|---|
| Ours (non-documentary) | 4.50 | 4.32 | 4.41 |
| Ours (documentary) | 4.50 | 4.17 | 4.33 |

## J    FAILURE CASES

**Consistency with occluded or unseen parts.** Consistency across shots can still be challenging, especially when reference images contain occluded or unseen parts Figure 28-(a).

**Inter-scene camera work transition.** While our method models multi-shot camera language within a scene, inter-scene camera transitions remain unexplored Figure 28-(b). Achieving smooth transi-

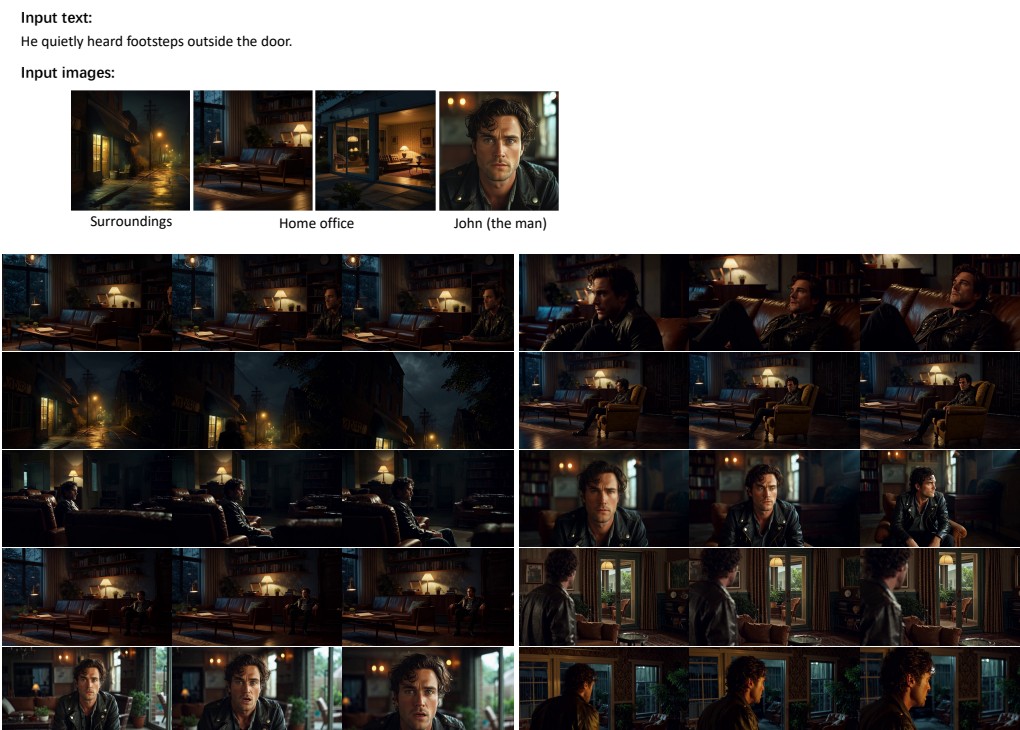

Figure 16: Qualitative results of the final generated film for the short input text of "He heard footsteps".

tions requires the MLLM to plan shared visual elements across scenes, which we leave for future work.

**Large-motion distortion from the base video model.** For scenes involving fast or complex motion, the underlying video generator may produce distortions. These artifacts are inherited from the base model and remain an open challenge.

## K    LIMITATIONS

While FilMaster represents a significant step towards automated professional film generation, it currently has certain limitations. 1) It inherits the limitations of current (M)LLMs. For example, current MLLMs operate at a relatively coarse temporal granularity (at the level of "second"), while more precise sub-second alignment is required for foley and sound effects design. This may cause audio-video mismatched issues. 2) Dialogue-heavy scenes are challenging, and our current system does not perform well here. There are two main reasons: (a) Shot/reverse-shot structure: Dialogue scenes require stable spatial layout (e.g., speaker A always on the left, speaker B on the right across cuts). This requires strong character-position tracking, which our system does not yet model. (b) Lip synchronization: Accurate lip-sync is essential in dialogue-heavy content but is not the primary focus of our method. 3) Advanced post-production techniques such as color grading and a diverse range of cinematic transitions are not yet incorporated into our system. These aspects, crucial for achieving a fully polished cinematic look and feel, were beyond the primary scope of this work, which focused on foundational camera language design and cinematic rhythm control. We acknowledge their importance and plan to address their integration in future research.

## L    USE OF LLMS

We used LLMs to improve the grammar, spelling, and clarity of our writing. The authors reviewed and edited all text and are responsible for the final content.

**LLM Prompt for Script Elaboration**

**Overall System Prompt:** *You are a professional director. You are going to write a script from 4 steps (1. Synopsis, 2. Simplified Storyboard, 3. Detailed Storyboard, 4. Scene) hierarchically. Never use Markdown.*

**Step 1: Synopsis**
You are in the first step (1. Synopsis). Design a synopsis based on the user input. The synopsis must contain:
- **Theme:** The central idea of the story.
- **Characters:** 2-5 characters, each with a name and a concrete visual description.
- **Plot:** A summary of the story's main events.

> **Example for Step 1**
>
> **Input:** `He waits anxiously.`
> **LLM Output:**
> ```
> a. Theme: Depicting a man's anxious and complex state of
>    mind.
> b. Characters: Man, Woman (Wife)
> c. Plot: A man anxiously waits at the hospital, constantly
>    worrying about his wife's health. Every time a doctor
>    appears, he hopes for answers...
> ```

**Step 2: Simplified Storyboard**
You are in the second step (2. Simplified Storyboard. Design a simplified storyboard based on the synopsis, structured into four parts: Background, Cause, Development, and Outcome.

> **Example for Step 2**
>
> **Input (Output from Step 1):**
> ```
> a. Theme: Depicting a man's anxious and complex state of
>    mind...
> ```
> **LLM Output:**
> ```
> a. Background: He is a programmer. Due to long hours of
>    overtime...
> b. Cause: He nervously sits in the hospital waiting area,
>    gripping...
> c. Development: He stands at the door of his home, tightly
>    gripping...
> d. Outcome: The next morning, the man received a call from
>    the doctor...
> ```

Figure 17: The illustration of prompts used for script elaboration. The process starts with a high-level synopsis and progressively refines it through four major steps into a detailed, structured storyboard. Here the first 2 steps are shown.

---

**LLM Prompt for Script Elaboration**

**Step 3: Detailed Storyboard**
You are in the third step (3. Detailed Storyboard). Extend the simplified storyboard with detailed visual descriptions, ensuring each sentence describes a simple, concrete 5-second shot. The visual descriptive writing can include both real and imaginary scenes. Real scenes consist of primary shots (specific descriptions) and reverse shots (descriptions of the surrounding environment), while imaginary scenes include unrelated shots (metaphors).

> **Example for Step 3**
>
> **Input (Output from Step 2):**
> ```
> a. Background: He is a programmer...
> ```
> **LLM Output:**
> ```
> a:
>     i. In an office, he is programming, his eyes fixed on
>     the computer...
>     ii. His stomach growls loudly, and he winces in pain,
>     his hand...
> b:
>     i. In a corner of the hospital, he curls up on a faded
>     blue bench...
> ```

---

**Step 4: Scene**
This final step involves segmenting the detailed storyboard into distinct scenes with the same spatio-temporal contexts: Group prompts chronologically and by location, extracting metadata (Time, Location, Characters, Visual Elements, Objective), adding detailed audio descriptions (voice-over, background ambiance, musical scoring) to each scene as placeholder.

> **Example for Step 4**
>
> **Input (Output from Step 3).**
> **LLM Output:**
> ```
> a. Time 1:
>     Space 1:
>         Summary: Evening, office, Man (focused, working,
>         programming on a computer).
>         Prompt:
>         (1) A man sits in a dimly lit office. His face is
>         illuminated by the blue glow of a computer screen.
>         His fingers type rhythmically on the keyboard.
>         (2) The man suddenly winces, clutching his stomach.
>         He leans back, rubbing his temples as the room
>         seems to spin.
>         Time: Evening; Location: Office; Character: Man;
>         Visual Elements: None;
>         Objective: Emphasize the man's daily routine of
>         overworking.
>         Audio:
>             (1) voice-over: (None)
>             (2) background ambiance: Quiet hum of a comp-
>             uter fan, distant city traffic.
>             (3) music: Low, monotonous synth track sugg-
>             esting routine.
> ```

Figure 18: The illustration of prompts used for script elaboration. The process shows the detailed storyboard and the final output, which is organized into scenes with spatio-temporal contexts.

**LLM Prompt in Multi-shot Synergized Camera Language Design**

The following instructions are provided to the LLM to enrich a given shot description with cinematic details based on retrieved examples.

```
Your role is to add the camera movements and the atmosphere for
each prompt.
Leave the "Time, Location, Character, Visual Elements, Object-
ive" intact. You only need to revise the "Prompt" part.

You can retrieve the similar prompts in the dataset, and design
suitable camera movements
and atmosphere that correspond to the scene.
Use similar templates as in the retrieved prompts to rewrite
the prompts.

Think step by step, and check your answers.
Never use Markdown.

Here is the user prompt:
{user_prompt}
```

**Note:** The placeholder {user_prompt} is dynamically replaced with the textual description of the scene.

Figure 19: The illustration of prompts used for retrieval and shot re-planning after spatio-temporal-aware indexing in our Multi-shot Synergized Camera Language Design module. The LLM is instructed to act as a cinematographer, using retrieved examples from a film dataset as a reference to rewrite shot prompts within the scene with professional camera language and atmospheric details.

---

**MLLM Prompt for Audience-Aware Review**

The MLLM is instructed to act as a simulated audience with a specific persona. This prompt enables generating the critical feedback that drives the refinement from *Rough Cut* to *Fine Cut*.

```
You are a movie viewer, with these demographic profile:
{demographic_profile}.
You need to evaluate the biggest problem of the video in the
aspect of video structure, rhythm, scene transitions, and audio.

Here is the script:
{script}

Here is the audio:
{audio}

Pay attention that the main idea and plots in script should not
be changed, only the representation in structure, rhythm, scene
transitions, and audio can be changed.
```

---

**Placeholders:**
- {demographic_profile}: A description of the target audience's characteristics and viewing preferences, which is generated by an LLM leveraging search tools. Here we use "short-drama audience".
- {script}: The input text.
- {audio}: The initial placeholder audio descriptions (VO) that accompany the video clips.

Figure 20: The illustration of prompts used to generate audience-aware feedback. The MLLM embodies a specific audience persona to critique a rough cut, focusing on structural and rhythmic elements rather than the core plot. This feedback is the primary input for the subsequent video editing and sound design.

---

**Prompting Pipeline for the Video Editing**

The video editing process is guided by a sequence of prompts. First, an MLLM generates a detailed, time-coded description of the video. This description then serves as a key input for two parallel LLM tasks that plan structural and durational adjustments.

**Step A: MLLM-based Video Description Generation**
This prompt instructs an MLLM to create a detailed, time-coded caption for each 5-second shot, providing a structured representation of the video content.

```
Describe the video based on the script, audio, and the
video.

Here is the script of the video:
{script}

Here is the audio of the video:
{audio}

For each scene (5s) in this video, generate detailed cap-
tions that describe the scene along with any spoken text
placed in quotation marks. Place each caption into an
object with the timecode of the caption.

Output example:
{{
"timecode": "00:00-00:05",
"caption": "A dirt path winds through a sunlit forest..."
}},
...
```

Figure 21: The prompting pipeline for the Video Editing. **(A)** An MLLM first annotates the video with detailed, time-coded descriptions. **(B)** This structured description, along with the audience analysis, is then fed into two parallel LLM prompts that generate plans for **Structure Adjustment** and **Duration Adjustment**, forming the complete editing plan for the *Fine Cut*.

---

**Prompting Pipeline for the Video Editing**

**Step B: LLM-based Editing Plan Generation**
Using the MLLM's output ({`video_description`}) and the audience analysis ({`analysis`}), two parallel prompts generate the final editing plan.

**Editing Task 1: Structure Adjustment**

```
You are good at structure adjustment for the video. The
main idea provided by the audio should be kept. For each
shot (5s) in this video, help me to reorder the shots ba-
sed on the video and the analysis.

Here is the description of the video:
{video_description}

Here is the audio for the video (each paragraph is a shot):
{audio}

Here is the analysis:
{analysis}

Requirements:
(1) Split the shots into scenes.
(2) Construct the structure of the video in the format of
timeline.
(3) Reorder the structure based on the analysis.
```

Figure 22: The prompting pipeline for the Video Editing. **(A)** An MLLM first annotates the video with detailed, time-coded descriptions. **(B)** This structured description, along with the audience analysis, is then fed into two parallel LLM prompts that generate plans for **Structure Adjustment** and **Duration Adjustment**, forming the complete editing plan for the *Fine Cut*.

**Prompting Pipeline for the Video Editing**

**Editing Task 2: Duration Adjustment**

```
You are good at time duration adjustment for the video.
For each shot (5s) in this video, numerate the shot, and
help me to suggest
the time duration (<=5s) for each shot based on the video
and the analysis, which enables the overall pacing.

Here is the description of the video:
{video_description}

Here is the analysis:
{analysis}

You need to determine the duration of each shot (5s), whe-
ther the video should be cropped directly, accelerated to
the suggested time duration, or retain the original dura-
tion (5s) to maintain overall pacing. If the suggested
duration is 5s, the mode is "retain".
If less than 5s, choose "crop" or "accelerate":
    -"Crop" is used for cutting down tedious information.
    -"Accelerate" is used for speeding up the video for
    pacing.

Check your answer and give the reasons.

Output example:
[
{{
"shot 1": "00:00-00:05",
"duration": "suggested time duration in second",
"mode": "crop, accelerate, or retain",
"reason": "xxx"
}},
...
]
```

Figure 23: The prompting pipeline for the Video Editing. **(A)** An MLLM first annotates the video with detailed, time-coded descriptions. **(B)** This structured description, along with the audience analysis, is then fed into two parallel LLM prompts that generate plans for **Structure Adjustment** and **Duration Adjustment**, forming the complete editing plan for the *Fine Cut*.

---

**Prompting Pipeline for Multi-Scale Sound Design**

The sound design process employs a multi-scale strategy, using distinct prompts to generate different audio layers at their appropriate temporal granularity.

This prompt instructs an LLM to generate the vocal track for the film, operating at the shot level (e.g., 5-second segments).

**Task 1: Shot-Level Sound Design**

```
You are good at adding audio (narration and dialogue) for
the video. For this video, help me to add narration or
dialogue based on the video and the analysis. You can
group shots into scenes if they describe the same thing.

Text Requirements:
- Narration and dialogue should not appear simultaneously.
- The total word number should be controlled to 1 word per
  second.
- Check the word number: Ensure the total number of words
  is appropriate for the time frame (e.g., <= 10 words for
  a 10s scene).

Here is the script of the video:
{script_simple}

Here is the description of the video:
{video_description}

Here is the analysis:
{analysis}

Output example:
[
{{
"time": "00:00-00:05",
"narration": "xxx",
"dialogue": "character: xxx",
"word number": xxx
}},
...
]
```

Figure 24: The prompting pipeline for our multi-scale Sound Design module. The process is split into two parallel tasks operating at different temporal scales. (**Task 1**) An LLM generates shot-level narration and dialogue, adhering to strict length constraints. (**Task 2**) An MLLM generates intra-shot, fine-grained foley and sound effects synchronized with specific visual actions. Together, these prompts create a rich, multi-layered soundscape.

**Prompting Pipeline for Multi-Scale Sound Design**

This prompt instructs an MLLM to generate fine-grained foley and sound effects by analyzing the video content on a second-by-second basis.

**Task 2: Intra-Shot Sound Design**

```
Here is the caption of the video: {video_caption}
Please describe the actions in this video frame by frame
with the timecode range. Give the suitable foley and sou-
ndeffect to the
frames described.

Output example:
[
{{
"00:00-00:01": "The camera follows the character from be-
hind as they begin to walk along a path into the forest."
"Foley": "Footsteps on a soft forest path with occasional
crunching sounds. Gentle rustling of leaves."
"Soundeffect": "Light magical twinkle or chime to empha-
size the forest ambiance"
}},
{{
"00:01-00:02": xxx
}},
...
]
Answer in json directly.
```

Figure 25: The prompting pipeline for our multi-scale Sound Design module. The process is split into two parallel tasks operating at different temporal scales. **(Task 1)** An LLM generates shot-level narration and dialogue, adhering to strict length constraints. **(Task 2)** An MLLM generates intra-shot, fine-grained foley and sound effects synchronized with specific visual actions. Together, these prompts create a rich, multi-layered soundscape.

**Objective**: Emphasize the **insignificance** of the Little Prince and the White Fox against the backdrop of the universe.

Prompt Summary: Morning, asteroid, Little Prince and White Fox standing quietly with wide eyes, stars twinkling.

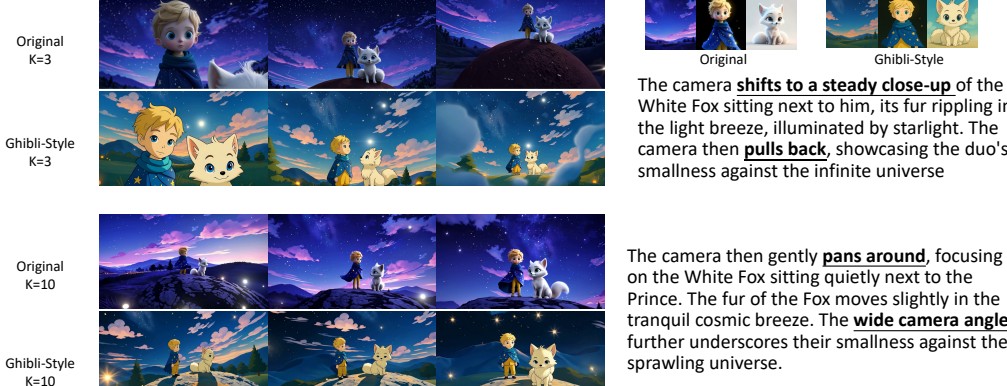

The camera **shifts to a steady close-up** of the White Fox sitting next to him, its fur rippling in the light breeze, illuminated by starlight. The camera then **pulls back**, showcasing the duo's smallness against the infinite universe

The camera then gently **pans around**, focusing on the White Fox sitting quietly next to the Prince. The fur of the Fox moves slightly in the tranquil cosmic breeze. The **wide camera angle** further underscores their smallness against the sprawling universe.

Figure 26: Varied cinematic style examples. We show our method can generate varied cinematic style videos by (1) changing the styles of reference images; (2) increase the retrieved number K.

**Input text:**

A documentary look at life in the desert and how living things survive the extreme environment.

**Input images:**

Desert    A local biologist    A wildlife photographer    A Bedouin elder with a lifetime of stories    A passionate botanist

Figure 27: **Our method can generalize to unseen domains**, such as documentary genre. Given the input text "A documentary look at life in the desert and how living things survive the extreme environment," our method first elaborates a script featuring relevant characters serving as observers, such as a local biologist, a wildlife photographer, a storytelling Bedouin elder, and a passionate botanist. Subsequently, it generates videos with camera language and cinematic rhythm.

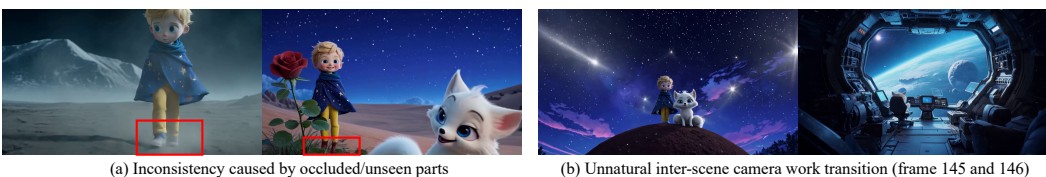

(a) Inconsistency caused by occluded/unseen parts      (b) Unnatural inter-scene camera work transition (frame 145 and 146)

Figure 28: **Failure cases of FilMaster.** (a) Inconsistency caused by occluded or unseen parts in reference images. Here, the color of the Little Prince's shoes changes from white to brown. (b) Unnatural inter-scene camera work transition. The visual focus shifts abruptly from the center in frame 145 to the left (White Fox) in frame 146, resulting in a jarring discontinuity.

