# OpenReview forum: "FilMaster: Bridging Cinematic Principles and Generative AI for Automated Film Generation"
_ICLR.cc/2026/Conference — ICLR 2026 Poster_

### Official Review · Reviewer_399a · 2025-10-17

**Soundness:** 3
**Presentation:** 3
**Contribution:** 3
**Rating:** 6
**Confidence:** 4

**Summary:**

The paper proposes FilMaster, an automated system for film generation that integrates cinematic principles—specifically, camera language design and cinematic rhythm control—into the video generation pipeline.
It introduces:

* A Multi-shot Synergized Camera Language Design module using a scene-level Retrieval-Augmented Generation (RAG) from a 440k film dataset to plan expressive, coherent shots.

* An Audience-Aware Cinematic Rhythm Control module that simulates post-production editing with “audience feedback” and multi-track sound design.

**Strengths:**

* Expressive camera work: Produces more natural and film-like results; shows improvement in “camera language.”

* Comprehensive multi-stage pipeline: Covers both pre- and post-production, integrating video, audio, and editing.

* Quantitative and qualitative validation: Outperforms baselines on FilmEval and user studies.

**Weaknesses:**

* Pipeline complexity: The system is overly intricate, requiring multiple LLMs and retrieval steps; real-time practicality is unclear.

* Manual intervention unclear: The amount of human selection or post-curation is not specified—e.g., how many outputs are generated and chosen.

* Failure cases not discussed: The paper lacks analysis of when and why the method fails (e.g., scene inconsistency, unnatural transitions).

* Audio-visual mismatch: Lip sync and timing issues occasionally appear, affecting realism.

* Aesthetic awkwardness: Some close-up (“in-your-face”) camera shots feel forced or unnatural compared to simpler long-shots like in LTX-Video.

**Questions:**

* Could modules (e.g., video generation, editing) be replaced with other backbones besides Kling? How modular is the system?

* What level of human intervention or curation is required—how many outputs are generated per scene, and how is the best one selected?

* What are the typical failure cases (e.g., incoherent motion, desynced audio, unrealistic rhythm)?

* Have you evaluated the generalization to unseen domains (e.g., documentaries, dialogue-heavy content)?

---

> ### Author Response · Authors · 2025-11-21
> **Response to Reviewer 399a [Part 1/3]**
>
> We sincerely thank the reviewer for your insightful comments and recognitions to this work, especially for acknowledging that:
>
> * Our work demonstrates expressive camera work: Produces more natural and film-like results; shows improvement in "camera language".
> * Our work presents comprehensive multi-stage pipeline: Covers both pre- and post-production, integrating video, audio, and editing.
> * Our method shows effectiveness through quantitative and qualitative validation: Outperforms baselines on FilmEval and user studies.
>
>
> We have polished the paper, added the experiment results and make the clarifications in the revised version (marked in blue). Note that the following polishments have been made according to your advice:
>
> * Failure cases are included in Appendix J "Failure Cases".
> * The generated results of lip-sync are included in Appendix F.6 "Results of Lip-Synchronization".
> * Results replaced with other backbones are included in Appendix F.5 "Using Open-Source Models".
> * Generalization to unseen domains (e.g., documentaries) is included in Appendix I "Generalization to Unseen Domains".
> * Audio-visual mismatch analysis and dialogue-heavy content is included in Appendix K "Limitations".
>
> Thanks again for your very constructive comments, which have helped us improve the paper quality significantly! Below we would like to provide point-to-point responses to all the raised questions:
>
>
>
> > **Q1: Pipeline complexity: real-time practicality.**
>
> **A1:** **1)** The pipeline mirrors real film production. Our design intentionally follows the standard filmmaking workflow, pre-production, production, and post-production. In this process, different roles (director, cinematographer, editor, sound designer) naturally require different types of reasoning. Thus we use multiple LLMs to simulate the process.
>
> **2)** Multiple LLMs improve robustness. Removing or merging modules increases hallucinations and errors. We include ablation studies in Appendix D, showing that the multi-LLM structure is more stable than a single large model handling all tasks at once.
>
> **3)** Practicality and runtime are comparable to existing systems. We compare the time cost of our pipeline with previous multi-LLM filmmaking systems. For post-production, since no prior work provides comprehensive automatic post-production, including structural adjustment, duration adjustment, and multi-layer sound design, we compare our system with human editors.
> Our added generation-time overhead is modest, and our automatic post-production is far faster than human editing. This shows that, despite the multi-step pipeline, the system is efficient.
>
> | Method       | LLM in generation (s/shot) ↓ | (M)LLM in post-production (s/shot) ↓ |
> | ------------ | ---------------------------- | ------------------------------------ |
> | MovieAgent   | 43.19                        | -                                    |
> | AnimDirector | 13.41                        | -                                    |
> | Human[A]     | -                            | 1380                                 |
> | Ours         | 16.15                        | 90.91                                |
>
>
>
>
>
> > **Q2: Manual intervention; What level of human intervention or curation is required—how many outputs are generated per scene, and how is the best one selected?**
>
> **A2:** **1)** Our system is fully automatic and requires minimal human curation.
> For each scene, we normally generate one output. We do not sample multiple versions for manual selection. The pipeline is designed to run end-to-end without human intervention.
>
> **2)** Automatic review during post-production.
> The post-production stage acts as an automatic second check. If a generated shot clearly deviates from the plot (e.g., distorted content or irrelevant elements), the structural organization step will automatically remove it. This reduces the need for manual filtering.
>
> **3)** Optional human editing interface.
> If users want further adjustments, they can optionally edit the outputs. We provide OTIO files that can be directly imported into standard editing software (e.g., DaVinci Resolve). However, this is not required for our main results.

---

> ### Author Response · Authors · 2025-11-21
> **Response to Reviewer 399a [Part 2/3]**
>
> > **Q3: Failure cases**
>
> **A3:**  Thank you for pointing out this missing part. We have added analysis of failure cases in Appendix J, and summarize key points as follows:
>
> **1)** Scene/ID consistency with occluded parts
> Consistency across shots can still be challenging, especially when reference images contain occluded or unseen parts.
>
> **2)** Inter-scene camera work transition.
> While our method models multi-shot camera language within a scene, inter-scene camera transitions remain unexplored. Achieving smooth transitions requires the MLLM to plan shared visual elements across scenes, which we leave for future work.
>
> **3)** Large-motion distortion from the base video model.
> For scenes involving fast or complex motion, the underlying video generator may produce distortions. These artifacts are inherited from the base model and remain an open challenge.
>
>
>
> > **Q4: Audio-visual mismatch: Lip sync and timing issues occasionally appear, affecting realism.**
>
> **A4:**  **1)** Audio-visual mismatch:
>
> * A few unsynchronized sound events are caused by minor inaccuracies in the input textual descriptions or asset timing annotations. Our method uses these inputs to determine when a sound should occur, so imperfect inputs may lead to slight offsets.
> * In addition, precise sub-second alignment remains challenging because current MLLMs operate at a relatively coarse temporal granularity. We have added this to the limitation section and will explore tighter temporal conditioning in future work.
>
> **2)** Lip sync and timing:
> Our method focuses on sound design and does not integrate a lip-sync module. Lip-sync is an orthogonal component and can be easily applied in post-processing. To demonstrate compatibility, we include lip-synchronized results in Appendix F.6 (Results of Lip-Synchronization) using an off-the-shelf lip-sync model, showing that this issue can be addressed within our pipeline.
>
>
>
> > **Q5: Aesthetic awkwardness: Some close-up (“in-your-face”) camera shots feel forced or unnatural compared to simpler long-shots like in LTX-Video.**
>
> **A5:**  **1)** We understand the reviewer’s concern. The close-up shots in our results are not random or forced; they are chosen to serve the narrative objective. Our camera module plans shots based on what the scene is trying to express.
>
> **2)** For example, in Figure 1, the camera intentionally moves in for a close-up of the Little Prince, the White Fox, and the Rose to highlight their emotional interaction. This is a cinematic choice when the goal is to show subtle expressions or relationships.
>
> In contrast, methods such as LTX-Studio mainly produce static or slowly moving long shots with characters centered in the frame. While these shots are visually clean, they often miss the intended narrative focus and cannot emphasize character interaction as effectively.
>
> > **Q6: Could modules (e.g., video generation, editing) be replaced with other backbones besides Kling? How modular is the system?**
>
> **A6:** Thank you so much for the precious advice! We have added analysis analysis of replacing the backbones in Appendix F.5. Here we summarize as follows:
>
> **1)** Our method is fully modular and can swap all major components (LLM, MLLM, video generator) with open-source models.
>
> **2)** GPT-4o --> DeepSeek-v3, LLaMA-3 70B
> We provided quantitative comparison in Table 12 in Appendix F.5,
> to test the effect of different LLMs for RAG. Across these different LLMs, our RAG-enhanced method maintains a strong performance, not depending on GPT-4o, achieving an average preference improvement of **55%** over a non-RAG baseline. This demonstrates the generalizability and robustness of our proposed method.
>
> **3)** Gemini --> Qwen-VL
> We replace Gemini with Qwen-VL. The results below show consistent performance gains over existing baselines, demonstrating that the system remains effective with open-source multimodal models.
>
> **4)** Kling video model --> Wan2.1-VACE[B]
> We replace Kling video model with Wan2.1-VACE model. Our method still outperforms MovieAgent, AnimDirector, and LTX-Studio, as shown below:
>
>
> | Method                 | CL (↑)   | CRh (↑)  | Avg (↑)  |
> | ---------------------- | -------- | -------- | -------- |
> | MovieAgent             | 2.96     | 1.94     | 2.45     |
> | AnimDirector           | 2.74     | 1.74     | 2.24     |
> | LTX-Studio             | 3.74     | 3.62     | 3.68     |
> | **Ours (Wan2.1-VACE)** | **4.11** | **3.94** | **4.03** |
> | **Ours (Qwen-VL)**     | **4.42** | **4.33** | **4.38** |

---

> ### Author Response · Authors · 2025-11-21
> **Response to Reviewer 399a [Part 3/3]**
>
> > **Q7: Have you evaluated the generalization to unseen domains (e.g., documentaries, dialogue-heavy content)?**
>
> **A7:** Thank you for pointing out this interesting question.
>
> **1)** Generalization to unseen domains (e.g., documentaries).
> Our method can generalize to unseen domains, such as documentaries. We provide qualitative examples in Appendix I, Figure 27, and quantitative as follows. The results show that our camera planning and post-production modules still function well in the setting of documentary.
>
> |                        | CL (↑) | CRh (↑) | Avg (↑) |
> | ---------------------- | ------ | ------- | ------- |
> | Ours (non-documentary) | 4.50   | 4.32    | 4.41    |
> | Ours (documentary)     | 4.50   | 4.17    | 4.33    |
>
> **2)** Dialogue-heavy content.
> Dialogue-heavy scenes are indeed challenging, and our current system does not perform well here. There are two main reasons:
>
> (a) Shot/reverse-shot structure:
> Dialogue scenes require stable spatial layout (e.g., speaker A always on the left, speaker B on the right across cuts). This requires strong character-position tracking, which our system does not yet model.
>
> (b) Lip synchronization:
> Accurate lip-sync is essential in dialogue-heavy content but is not the focus of our method.
>
> We discuss dialogue-heavy storytelling as an important direction in Appendix K, and will explore it in future work.
>
>
> ---------
>
> **References**
>
> [A] StockMusic.net. (n.d.). *How long does post-production take?* StockMusic.net. https://stockmusic.net/blog/how-long-does-post-production-take/
>
> [B] Jiang Z, Han Z, Mao C, et al. Vace: All-in-one video creation and editing[J]. arXiv preprint arXiv:2503.07598, 2025.

---

### Official Review · Reviewer_rwQb · 2025-10-20

**Soundness:** 3
**Presentation:** 3
**Contribution:** 3
**Rating:** 6
**Confidence:** 5

**Summary:**

This paper presents a fully automated, end-to-end clip-level video generation agent. By providing multiple reference images and an initial prompt as input, the system invokes different models to iteratively modify both the prompt and the video. During the Coordination stage, the agent also retrieves suitable audio from an audio library for adaptation. The construction of cinematic language, agent-level video editing (such as acceleration and scene switching), and audio synchronization are particularly novel aspects of the approach. The experimental results, as well as the videos included in the supplementary materials, are impressive and refreshing.

**Strengths:**

1. An automatic end-to-end clip-level video generation agent with impressive performance.
2. The Coordination Stage could edit the order and duration of the generated videos, which is reasonable as inter-clip videos do not have strict temporal order constraint. The audio fusion manner is also intuitive and suitable for agent like methods (retrieval and synchronized).
3. The whole paper is well writen and easy to understand.

**Weaknesses:**

The performance of this paper is great, my main concern lies on fair comparison with existing methods. This paper utilizes Kling 1.6 as the video generation model, which is much better than existing open-source model that previous method used, such as CogVideoX and LTX-Video. So is the performance improvement simply caused by the basic ability of Kling? It is recommended to have a fair comparison with MovieAgent to see the contribution of this paper.

**Questions:**

See the Weaknesses.

---

> ### Author Response · Authors · 2025-11-21
> **Response to Reviewer rwQb**
>
> We sincerely thank the reviewer for your insightful comments and recognitions to this work, especially for acknowledging that:
>
> * Our automatic end-to-end video generation agent with impressive performance;
> * The reasonable design of Coordination Stage that could edit the order and duration;
> * The retrieval and synchronized audio fusion manner that is intuitive and suitable;
> * The whole paper is well writen and easy to understand.
>
> Thanks again for your very constructive comments, which have helped us improve the paper quality significantly! Below we would like to provide point-to-point responses to the raised question:
>
>
>
> > **Q1: Fair comparison with existing methods.**
>
> **A1:** **1) We provide fair comparisons using the same base model (Kling 1.6).**
> To ensure fairness, we re-implemented MovieAgent and Anim-Director on top of the same video backbone, Kling 1.6. Quantitative results are reported in Table 3 and Table 7, and qualitative examples are shown in Appendix Figure 12. We report the results as follows:
>
> User study (all methods using Kling 1.6):
>
> | Method              | NS ↑     | AT ↑     | AE ↑     | RF ↑     | EE ↑     | OE ↑     | CL ↑     | CRh ↑    | Avg ↑    |
> | ------------------- | -------- | -------- | -------- | -------- | -------- | -------- | -------- | -------- | -------- |
> | Anim-Director-Kling | 1.94     | 2.35     | 1.44     | 1.94     | 1.84     | 2.20     | 2.16     | 1.85     | 1.95     |
> | MovieAgent-Kling    | 1.32     | 2.38     | 1.68     | 2.02     | 1.96     | 1.92     | 2.01     | 1.89     | 1.88     |
> | **Ours-Kling**      | **3.70** | **3.80** | **3.80** | **3.73** | **3.93** | **3.87** | **3.76** | **3.82** | **3.79** |
>
> Automatic evaluation (all using Kling 1.6):
>
> | Method              | NS ↑     | AT ↑     | AE ↑     | RF ↑     | EE ↑     | OE ↑     | CL ↑     | CRh ↑    | Avg ↑    |
> | ------------------- | -------- | -------- | -------- | -------- | -------- | -------- | -------- | -------- | -------- |
> | Anim-Director-Kling | 2.10     | 3.30     | 2.70     | 2.50     | 2.20     | 2.40     | 3.02     | 2.38     | 2.70     |
> | MovieAgent-Kling    | 1.50     | 2.95     | 2.30     | 1.90     | 2.00     | 2.00     | 2.55     | 1.98     | 2.27     |
> | **Ours-Kling**      | **4.25** | **4.55** | **4.10** | **4.70** | **4.20** | **4.40** | **4.50** | **4.32** | **4.41** |
>
> **2) The performance gain is not simply from Kling’s stronger generation ability.**
> Kling mainly improves visual fidelity and motion smoothness, but it does not provide camera-language planning, multi-shot synergy, scene-level coherence, narrative rhythm or pacing, audience-aware editing. These abilities are essential for filmmaking but not provided by the base model.
>
> **3) Our improvements come from our method, not the backbone.**
> Our method adds two key components that baseline methods do not have: Multi-shot synergized RAG for scene-level camera design aligned with narrative objectives, and generative post-production that adjusts structure, pacing, rhythm, and audio. These modules improve film quality beyond visual appearance, which explains why even with the same Kling model, our system notably outperforms MovieAgent-Kling and Anim-Director-Kling.

---

### Official Review · Reviewer_uN4o · 2025-10-20

**Soundness:** 2
**Presentation:** 1
**Contribution:** 1
**Rating:** 2
**Confidence:** 5

**Summary:**

This paper introduces FilMaster, an end-to-end automated film generation system designed to address key limitations in existing AI-based video generation. FilMaster proposes a Multi-shot Synergized Camera Language Design module which enables scene-level RAG that learns cinematography from movie references. To achieve a professional narrative, FilMaster further introduces Audience-Aware Cinematic Rhythm Control module for post-production workflow. The visualization experiments provide qualitative evidence for the effectiveness of the proposed method.

**Strengths:**

- The overall method is easy to follow.
- The visualizations in the supplementary materials offer clear qualitative support for the method's effectiveness.

**Weaknesses:**

**1. Limited Novelty**

I do not typically raise concerns about novelty lightly, but I must state that the technical contribution of this paper is highly limited. At its core, the proposed method is a relatively straightforward application of RAG. Crucial generative capabilities, such as identity preservation and high-fidelity video synthesis, appear to be inherited from the underlying foundational models used, rather than being novel contributions of the FilMaster framework itself. While the academic community certainly welcomes simple yet effective methods, such contributions usually offer a fundamental insight into the problem being studied. I do not believe this paper achieves that. Instead of solving a core technical challenge, the work primarily focuses on orchestrating existing components. For these reasons, the paper reads more like a well-executed technical project than a piece of novel research.

**2. Subjective Evaluation**

The primary evaluation is based on a user study, which, while valuable, is inherently subjective. Although the authors supplement this with an automatic evaluation using Gemini, this approach is also a form of subjective assessment. The paper would be much stronger if it included more objective and popular metrics. For example, text-video similarity scores could be used to measure the faithfulness of script elaboration, and a quantitative metric for identity consistency across shots would provide more robust evidence of the system's capabilities.

I understand a potential reason why authors did not discuss these metrics might be that these objective metrics would primarily test the performance of the base video generation model, not the proposed framework. However, this argument circles back to the core issue of novelty. If the main contribution is limited to the design of using cinematic language, and the measurable technical improvements are attributable to the underlying models, then the novelty of the framework itself is insufficient.


**Conclusion**

The paper is methodologically sound, but its contribution feels incremental. It does not present significant shortcomings, but conversely, it lacks any clear and compelling advantages that would distinguish it from prior work. The contribution feels unsubstantial and does not appear to meet the bar for acceptance.

I‘m willing to reconsider my score if the authors can provide a convincing rebuttal that thoroughly addresses my concerns.

**Questions:**

Please See Weaknesses

**Details Of Ethics Concerns:**

The authors have commendably identified two significant ethical considerations for their work. First, there is the issue of societal biases from the source films. Second, the potential for misuse as a generative tool warrants serious consideration.

---

> ### Author Response · Authors · 2025-11-21
> **Response to Reviewer uN4o [Part 1/3]**
>
> We sincerely thank the reviewer for your insightful comments and recognitions to this work, especially for acknowledging that:
>
> * The overall method is easy to follow.
> * The visualizations in the supplementary materials offer clear qualitative support for the method's effectiveness.
>
> We have polished the paper, added the experiment results and make the clarifications in the revised version (marked in blue). Note that the following polishments have been made according to your advice:
>
> * The rewritten criteria to reduce subjectivity are included in Appendix E "Filmeval Evaluation Instructions".
>
> Thanks again for your very constructive comments, which have helped us improve the paper quality significantly! Below we would like to provide point-to-point responses to all the raised questions:
>
>
>
> > **Q1: Novelty and contribution**
>
> **A1:** Thank you for your question. We would like to point out our novelty, contributions, and the differences from existing works.
>
> **1)** We propose a novel system for end-to-end automated film generation that integrates real-world cinematic principles from script to editable films. To our best knowledge, we are the first to explore generation and post-production into one automatic film generation system that can pay attention to both **camera language and cinematic rhythm**. Theses parts are important in film expression[A,B], but remained *unexplore* in previous works [C,D,E,F].
>
> **2)** We propose a novel **scene-level Retrieval-Augmented Generation (RAG)** framework for Multi-shot Synergized Camera Language Design. This is motivated by filmmakers that learn their camera language skills by studying extensive film references. We would like to clarify that our work is ***not*** relatively straightforward application of RAG. Direct usage of RAG for each shot (***naive RAG***), which retrieves references independently, often leads to visual incoherence (Appendix F.5). Instead, we design a script generation for generating a *Scene*, which shares the same spatio-temporal contexts and a narrative objective (Appendix A), and then use the entire scene to retrieve similar contexts, and design the camera language simultaneously for all shots within one scene. This scene-level retrieval and camera design improves coherence between shots.
> We compare naive RAG and our proposed scene-level RAG qualitatively and quantitatively in Table 12 and Figure 8, and report results as follows:
>
> | Ablation on RAG                | Score ↑   |
> | ------------------------------ | --------- |
> | w/o RAG                        | 0.100     |
> | w/ RAG, Shot-Level (naive)     | 0.133     |
> | **w/ RAG, Scene-Level (Ours)** | **0.511** |
>
> **3)** We propose audience perspective in post-production stage. This is motivated by real film post-production that need coorporation among different professional roles (e.g., editors, sound designers). **Different from existing methods** that (a) generate and concatenate directly video clips, but do not model editing structure, temporal rhythm, or audiovisual design; (b) rely solely on the director’s perspective, which often leads to pacing issues or misalignment with audience expectations, we present audience-aware cinematic rhythm control. We (a) introduce multiple (M)LLMs for structural adjustment, timing and duration, and multi-layer sound design. We present robust-oriented mechanisms to boost robustness (Appendix D); (b) regard audience as important perspective for the generated outputs, which can effectively modify the organization formats to meet expectation of audience. We show the quantitative results in Table 12, and report as follows:
>
> | Ablation on Audience Dimensions | Score ↑   |
> | :------------------------------ | :-------- |
> | w/o Audience                    | 0.250     |
> | w/o Structure                   | 0.063     |
> | w/o Timing                      | 0.210     |
> | w/o Audio                       | 0.100     |
> | **Ours**                        | **0.380** |
>
> **4)** We would like to modestly point out that our work is not orchestrating existing components. Directly combining MLLMs together still lack the ability to accomplish the complex post-production (often hallucination, ineffective corrections). Instead, we design the following to boost the robustness of our system. **(a)** separating review of Rough Cut and video editing; **(b)** adding audience perspective to help edit the videos; **(c)** disentangle multimodal information for (M)LLMs for processing information. We add analysis in appendix D.

---

> ### Author Response · Authors · 2025-11-21
> **Response to Reviewer uN4o [Part 2/3]**
>
> **5)** We would like to clarify that our contributions are ***not*** from the underlying model that with crucial generative capabilities, such as ID preservation and high-fidelity video synthesis. **(a)** ID perservation, while important, it is not the primary focus of our work. Our primary focus is on the design of camera language and cinematic rhythm; **(b)** We compare the exisiting method with the same Kling 1.6 base model. Results below show that while strong base model show capability in high quality generation, existing film methods still lack the ability to design effective camera language and lack post-production stage, which leads to flat-pace outputs.
>
> | Method              | CL ↑     | CRh ↑    | Avg ↑    |
> | ------------------- | -------- | -------- | -------- |
> | Anim-Director-Kling | 3.02     | 2.38     | 2.70     |
> | MovieAgent-Kling    | 2.55     | 1.98     | 2.27     |
> | **Ours-Kling**      | **4.50** | **4.32** | **4.41** |
>
>
>
>
>
>
>
> > **Q2: Subjective Evaluation**
>
> **A2:** Thank you for your constructive suggestion.
>
> **1)** Using MLLM as evaluator is a common practice in diffusion text-to-image [G] and text-to-video [H, I, J] generation when no established automatic metric adequately captures multi-dimensional film quality (narrative, camera language, rhythm, etc.). Because existing metrics do not evaluate such holistic aspects, we design a multi-dimensional rubric with 12 sub-criteria, each tied with detailed properties.
>
> **2)** We **rewrite the criteria** in our evaluation to reduce subjectivity in Appendix E. For example, in measuring narrative coherence, we replace "1 point: Chaotic and disorderly story with serious logical contradictions and plot discontinuities" **with more observable metrics** "1 point: Chaotic and disorderly — exhibits ≥3 major logical problems (clear causal errors, contradictory temporal order, character actions contradict prior information, or plot discontinuities) that prevent coherent understanding.". We provide the latest results as follows. The evaluation results below also support the effectiveness of our method.
>
> | Method             | CL ↑     | CRh ↑    | Avg ↑    |
> | ------------------ | -------- | -------- | -------- |
> | MovieAgent         | 2.26     | 1.54     | 1.90     |
> | MovieAgent-Kling   | 2.35     | 2.28     | 2.32     |
> | AnimDirector       | 2.60     | 1.70     | 2.15     |
> | AnimDirector-Kling | 2.60     | 2.30     | 2.45     |
> | LTX-Studio         | 3.35     | 3.23     | 3.29     |
> | **Ours-Kling**     | **4.10** | **3.99** | **4.04** |
>
> **3)** We measure the stability of Gemini as automatic evaluation, by conducting 1/3/5/10 repeated evaluations. The results below show extremely low variance (CV 0.31%–2.99%), indicating that automatic scoring is highly stable and not sensitive to randomness.
>
> | Method              | 1    | 3    | 5    | 10   | Standard Deviation | Coefficient of Variation(CV) |
> | ------------------- | ---- | ---- | ---- | ---- | ------------------ | ---------------------------- |
> | Anim-Director       | 2.15 | 2.13 | 2.11 | 2.12 | 0.0171             | **0.80%**                    |
> | Anim-Director-Kling | 2.45 | 2.49 | 2.43 | 2.51 | 0.0365             | **1.48%**                    |
> | MovieAgent          | 1.90 | 1.91 | 1.88 | 1.86 | 0.0222             | **1.17%**                    |
> | MovieAgent-Kling    | 2.32 | 2.21 | 2.18 | 2.18 | 0.0665             | **2.99%**                    |
> | LTX-Studio          | 3.29 | 3.27 | 3.27 | 3.27 | 0.01               | **0.31%**                    |
> | Ours-Kling          | 4.04 | 4.08 | 4.09 | 4.1  | 0.0263             | **0.64%**                    |
>
>
>
> **4)** We would like to clarify that existing text-video similarity or ID consistency metrics are not suitable in our setting. These metrics offer objective analysis but may not fully match human preferences, as also observed in [C]. Text-video similarity often uses CLIP-T, which calculates similarity between frames and text, fails for longer video when objects are not present at each frames. ID consistency commonly uses Dino and CLIP-I to assess subject fidelity. ID subject with larger part usually gains higer score.
>
> **5)** We respectfully disagree with the concern that improvements stem only from stronger backbone models. Our improvements are ***not*** relied on the underlying models.
> To address this, we perform: **(a)** Ablation using the same underlying video model. We compare "with vs. without our camera language module" using identical video generators. Results (Table 12, Fig. 8) show clear degradation without our camera language design. This demonstrates that our improvements arise from the framework’s design, not the backbone. **(b)** Comparison with prior works using the exact same Kling base model. Even under identical backbone conditions, our method significantly outperforms prior film-generation systems, confirming that our scene-level RAG, and audience-aware post-production contribute directly to the improvement.

---

> ### Author Response · Authors · 2025-11-21
> **Response to Reviewer uN4o [Part 3/3]**
>
> > **Q3: Differences from prior work**
>
> **A3:** Thanks for your constructive question. We would like further to clarify our differences from previous storytelling works, film generation works, and existing film generation commercial product.
>
> **1)** **Storytelling or multi-shot generation works** [K,L,M]:
> Prior storytelling systems mainly target ID consistency or semantic continuity, but they **do not model camera language or cinematic rhythm**. We introduce the design of camera language, including shot types, camera movements, angles, and atmospheric characteristics for the designed narrative objective. This enables expressive cinematic storytelling (e.g., "use a long dolly-out to emphasize loneliness"), which prior storytelling systems cannot achieve.
>
> **2) Film generation works** [C,D,F]:
>
> * Although prior work [C] includes camera-language design, it relies on an LLM to imagine shot types and movements, **without grounding in real cinematic references**. In contrast, our method uses a retrieval-based camera-language module, allowing the shot plan to be guided by actual film examples, making the camera design more coherent and narratively appropriate.
> * Prior film-generation systems stop at script writing and video generation **from a director-only perspective**. They lack any form of automatic post-production. Our work introduces a full generative post-production stage, including structural editing, duration adjustment, and multi-layer sound design. Moreover, we incorporate an audience-perspective reviewer to refine the Rough Cut and adjust the final expression format, which is an element not present in prior academic systems.
>
> **3)** **Film generation commercial product** [E]:
> Commercial systems offer convenient pipelines but still rely **heavily on manual post-production**. They lack both transparency and automated editing. Our system provides capabilities that commercial products do not automatically support: **(a)** Fully automatic post-production, including structural organization, temporal rhythm adjustment, and audience perspective guided refinement; **(b)** Comprehensive multi-layer sound design (5 types), instead of only 1–2 types in commercial systems; **(c)** Explicit camera-language-based planning, rather than latent black-box generation.
>
>
>
> -----------------
>
> **References**
>
> [A] Nelmes J. Introduction to film studies[M]. Routledge, 2012.
>
> [B] Case D. Film technology in post production[M]. Routledge, 2013.
>
> [C] Wu W, Zhu Z, Shou M Z. Automated movie generation via multi-agent cot planning[J]. arXiv preprint arXiv:2503.07314, 2025.
>
> [D] Xu Z, Wang J, Wang L, et al. Filmagent: Automating virtual film production through a multi-agent collaborative framework[M]//SIGGRAPH Asia 2024 Technical Communications. 2024: 1-4.
>
> [E] LTX Studio. LTX Studio. https://app.ltx.studio/, 2024. Accessed: 2025-04. Organization: Lightricks.
>
> [F] Li Y, Shi H, Hu B, et al. Anim-director: A large multimodal model powered agent for controllable animation video generation[C]//SIGGRAPH Asia 2024 Conference Papers. 2024: 1-11.
>
> [G] Wu C, Zheng P, Yan R, et al. OmniGen2: Exploration to Advanced Multimodal Generation[J]. arXiv preprint arXiv:2506.18871, 2025.
>
> [H] Li Y, Xia M, Liu G, et al. AdaViewPlanner: Adapting Video Diffusion Models for Viewpoint Planning in 4D Scenes[J]. arXiv preprint arXiv:2510.10670, 2025.
>
> [I] Sun K, Huang K, Liu X, et al. T2v-compbench: A comprehensive benchmark for compositional text-to-video generation[C]//Proceedings of the Computer Vision and Pattern Recognition Conference. 2025: 8406-8416.
>
> [J] Qiu L, Li Y, Ge Y, et al. AnimeShooter: A Multi-Shot Animation Dataset for Reference-Guided Video Generation[J]. arXiv preprint arXiv:2506.03126, 2025.
>
> [K] Long F, Qiu Z, Yao T, et al. Videostudio: Generating consistent-content and multi-scene videos[C]//European Conference on Computer Vision. Cham: Springer Nature Switzerland, 2024: 468-485.
>
> [L] Zhao C, Liu M, Wang W, et al. Moviedreamer: Hierarchical generation for coherent long visual sequence[J]. arXiv preprint arXiv:2407.16655, 2024.
>
> [M] Meng Y, Ouyang H, Yu Y, et al. HoloCine: Holistic Generation of Cinematic Multi-Shot Long Video Narratives[J]. arXiv preprint arXiv:2510.20822, 2025.

---

> ### Comment · Reviewer_uN4o · 2025-11-28
>
> I would like to thank the authors for their detailed response and the additional clarifications provided during the rebuttal period.
>
> As a researcher who has also invested significant effort in this field, I fully acknowledge the solid implementation and the empirical efforts demonstrated in this work.
>
> Actually, after carefully considering the rebuttal for days, my primary concern is only partly resolved: the paper reads more like a comprehensive technical report or an engineering project rather than a research paper. While the authors have successfully addressed several practical issues, the technical contributions appear somewhat incremental.
>
> That being said, I value the substantial effort and the framework the authors have built. To reflect my recognition of the authors' effort, I am willing to raise my rating to 6. But I will lower my confidence score, allowing the AC and other reviewers to weigh in more heavily if they think this work is sufficient for acceptance.
>
> Finally, I would like to suggest that the authors focus more on the critical pain points of the domain. Although the method provides valid improvements, the contribution may not offer the sufficient impact required to advance the field significantly. I hope the authors will tackle these more challenging problems in their future work. :)
>
>
> (It seems there is a system bug currently preventing reviewers from modifying scores. I will update as soon as the issue is resolved.)

---

> > ### Author Response · Authors · 2025-11-28
> > **Thanks for your comments!**
> >
> > Dear Reviewer uN4o,
> >
> > We are truly grateful for your time in revisiting our work and for raising your score. We deeply appreciate your acknowledgement of our rebuttal efforts, the solidity of our implementation, and the empirical value of the proposed framework.
> >
> > As you mentioned being a researcher deeply invested in this field, your recognition of the "solid implementation" and "empirical efforts" means a lot to us. We respectfully believe that systematic innovation is a crucial stepping stone for the current video generation community. By bridging the gap between foundational models and professional workflows, FilMaster aims to serve as a solid baseline and a practical tool to inspire future research into fully automated film generation.
> >
> > We also take your advice on focusing on "critical pain points" to heart. Your insight motivates us to delve deeper into the fundamental challenges of high-fidelity synthesis and narrative consistency in our future work.
> >
> > Thank you again for your constructive feedback and for taking the time to reassess our submission.
> >
> > Best regards,
> >
> > Paper 864 Authors

---

### Official Review · Reviewer_Pb14 · 2025-10-30

**Soundness:** 2
**Presentation:** 3
**Contribution:** 2
**Rating:** 4
**Confidence:** 4

**Summary:**

The paper aims at automated film video generation, i.e., automatically generating film videos given a text and sets of reference images for characters and locations as input. The proposed framework, named "FilMaster", attempts to surpass existing methods in terms of camera language and cinematic rhythm. The core contributions lie in the introduced Multi-shot Synergized Camera Language Design and Audience-aware Cinematic Rhythm Control, where the former is designed for camera language by introducing a shot-level RAG, and the latter module serves as a post-production process to refine the "Rough Cut" to "Fine Cut" for cinematic rhythm. FilMaster is evaluated on the proposed FilmEval benchmark and outperforms recent methods (Anim-Director, MovieAgent, and LTX-Studio) in both camera language and cinematic rhythm, which is further evidenced through a user study. The paper also presents ablation studies regarding the two proposed modules.

**Strengths:**

1. The paper tackles an important and underexplored area—bridging cinematic principles and AI-based film generation. The topic is both academically relevant and practically impactful.
2. The authors explicitly ground their work in film principles (camera language, cinematic rhythm, audience perception, etc.) and emulate professional filmmaking workflows. This fills an existing academic gap between generative modeling and film studies.
3. The scene-level retrieval and coordinated camera planning improve shot-level incoherence in previous RAG-based methods.

**Weaknesses:**

1. While the system design is well-engineered, it lacks a deep technical innovation or theoretical insight at the algorithmic level. I would prefer to see more technically substantive modules rather than a purely workflow-oriented system.
2. The system’s multi-stage pipeline (retrieval -> shot planning -> rough cut -> audience feedback -> fine cut -> sound production) introduces fragility. The paper doesn’t investigate how errors propagate or how robust the system is when upstream stages fail.
3. Despite providing reference images, generated scenes may still lack consistency across shots, especially for partially occluded or unseen parts of reference subjects/scenes. The paper does not discuss this, though it is crucial for multi-shot storytelling.
4. Some generated results in the supplementary video exhibit noticeable audio–video misalignment, including unsynchronized sound effects and inaccurate lip movements.
5. The workflow depends on GPT-4o, Gemini, and Kling video generators. This makes reproducibility extremely difficult and limits transparency.

**Questions:**

It remains unclear whether the system can produce varied cinematic styles or if retrieval biases constrain creativity?

---

> ### Author Response · Authors · 2025-11-21
> **Response to Reviewer Pb14 [Part 1/3]**
>
> We sincerely thank the reviewer for your insightful comments and recognitions to this work, especially for acknowledging that:
>
> 1. Our work addresses an important and underexplored area (cinematic principles and AI-based film generation).
> 2. The paper bridges cinematic principles with AI film generation.
> 3. We ground our work in film principles and professional workflows.
> 4. Our scene-level retrieval and camera planning improve shot coherence.
>
>
> We have polished the paper, added the experiment results and make the clarifications in the revised version (marked in blue). Note that the following polishments have been made according to your advice:
>
> * The analysis of robustness is included in Appendix D "Robustness Analysis of Filmaster".
> * Discussion of consistency with partially occluded or unseen part is included in Appendix J "Failure Cases".
> * Sound effects issues are discussed in Appendix K "Limitations"
> * Generated results of lip-sync are included in Appendix F.6 "Results of Lip-Synchronization".
> * Results of replacing Gemini, Kling with open-source models are included in Appendix F.5 "Using Open-Source Models".
> * Results of varied cinematic styles are included in Appendix I "Varied Cinematic Styles".
>
> Thanks again for your very constructive comments, which have helped us improve the paper quality significantly! Below we would like to provide point-to-point responses to all the raised questions:
>
>
>
> > **Q1: Technical innovation or theoretical insight**
>
> **A1**: We thank the reviewer for the insightful comment. Our method introduces concrete technical innovations beyond engineering design. We summarize the key points below.
>
> **1)** End-to-end pipeline enabling professional editing (**system-level novelty**)
> Prior approaches either focus solely on generation [C,D,F] or still require substantial manual editing in practice [E]. Inspired by film production workflows in real world, we present the first pipeline that automatically spans script → generation → generative post-production → industry-standard NLE (non-linear editing) tools. This frames **film creation with camera language and rhythm** as a generative modeling problem rather than a workflow assembly.
>
> **2)** Multi-shot synergized RAG for scene-level camera planning (**algorithmic novelty**)
> Filmmakers learn camera language by studying film references. Motivated by this, we use retrieval of relevant film clips. However, *naïve per-shot RAG* produces fragmented, inconsistent camera work (Appendix F.5). To address this, we propose a **scene-structured RAG method** with:
>
> * Scene-level querying: retrieval is conditioned on the entire scene, not individual shots.
> * Multi-shot synergy: retrieved references provide shared spatio-temporal and narrative context.
> * Objective-driven camera planning: each scene has an explicit narrative objective guiding coherent camera design.
>   This goes beyond applying RAG and leads to coherent, intra-scene camera language, aligning more closely with real cinematography.
>
> **3)** Audience-aware generative post-production (**modeling innovation**)
> Existing methods generate and concatenate directly video clips,  but do not model editing structure, temporal rhythm, or audiovisual design. They also rely solely on the director’s perspective, which often leads to pacing issues or misalignment with audience expectations. Inspired by real film post-production that need coorporation among different professional roles (e.g., editors, sound designers), we introduce:
>
> * a coordination among (M)LLMs that adjusts structure, pacing, and multi-layer sound design for automatic post-production.
> * an **audience-aware perspective** that review the Rough Cut of video clips, providing an alternative perspective besides "director".
> * **separation** review of Rough Cut and video editing, **disentangling** multimodal information for (M)LLMs for processing information to further boost robustness of multiple (M)LLMs.
>
> This extends narrative modeling beyond "director-only generation" and brings generative systems closer to real film editing practices.
>
> **4)** Scope and Future Work
>
> Our work provides an attempt toward generative filmmaking. The integration of scene-level RAG camera language modeling, and audience-aware generative post-production represents technical progress beyond workflow integration. Components such as scene blocking (camera–actor interaction) remain open and will be explored in future work.

---

> ### Author Response · Authors · 2025-11-21
> **Response to Reviewer Pb14 [Part 2/3]**
>
> >  **Q2: Errors propagation and robustness**
>
> **A2:** We agree with the reviewer that a multi-stage pipeline can be fragile due to error propagation across stages. To address this, we intentionally design the system with multi-agent and several robustness-oriented mechanisms to mitigate failures. We also provide quantitative and qualitative robustness analysis in Appendix D. We summarize the key points below.
>
> **1)** We use **multi-agent** over single-agent pipelines to perform long-horizon filmmaking.
> Using a single agent to perform long-horizon filmmaking tasks causes compound reasoning overload. Our multi-agent, stage-decoupled architecture is intentionally designed to reduce error amplification by isolating perception, planning, and decision-making. This follows common practice in long-horizon agentic systems [A,B] and is more robust than single-agent approaches.
>
> **2)** We introduce three robustness-oriented design choices (Appendix D):
>
> * We **decouple** "review" and "correction" to prevent reasoning overload. Specifically, we separate review of Rough Cut and video editing. If we combine review and correction together into a single stage with an MLLM, it will be overwhelmed with the task, and be impacted the reasoning and analysis ability. Thus to keep both the analysis, correction ability, we separate the review of Rough Cut and video editing.
>
> * We add **audience-perspective** feedback to stabilize post-production. We observed that if we remove the perspective of audience, the post-production stage will be dominated by the perspective of director in the generation stage: There is no correction of the Rough Cut.
>
> * We **disentangle** multimodal perception and cognitive reasoning. Instead of feeding all multimodal inputs (video, audio, text script) into one model, our approach first uses an MLLM to generate detailed textual captions of the video. Then, a separate LLM uses these captions, alongside the script and placeholder audio descriptions, to perform the complex reasoning and editing. This separation of multimodal perception (captioning) from cognitive reasoning (editing) reduces complexity.
>
> **3)** Experiments of robustness analysis are provided in Appendix D.
> We include quantitative analysis by calculating correction ratio, success ratio. With our designs, our system is more robust in both correction and success ratio: (a) We found that by separating the clear task, the system will be more robust and have high scores in correction ratio and success ratio; (b) Adding audience perspective helps to effectively refine videos, with both higher scores in correction and success ratio; (c) The separation of multimodal perception (captioning) from cognitive reasoning (editing) reduces complexity, help increase the success ratio.
> We also include qualitative results showing failure recoveries.
>
>
>
> #
>
> > **Q3: Consistency across shots**
>
> **A3:** Thank you for pointing out this problem.
>
> **1)** We agree that cross-shot consistency is crucial for multi-shot storytelling. Our method does not explicitly cross-shot identity consistency, but several design choices help maintain it in most cases (Appendix A): **(a)** Coarse-to-fine script elaboration ensures that all shots are derived from a unified set of character descriptions. **(b)** Scene-level structuring groups multiple shots under the same scene, so they share the same characters, location, and narrative objective. These constraints naturally reduce inconsistency across shots.
>
> **2)** We also acknowledge that handling partially occluded or unseen parts of reference subjects/scenes is still a challenging problem for current generative models. Our method inherits these limitations. We include this as a failure case and discuss it in Appendix J. We will explore occluded or unseen identity/scene consistency in future work.
>
>
>
>
>
>
>
> > **Q4: Unsynchronized sound effects and inaccurate lip movements**
>
> **A4: 1)** Sound effects misalignment:
>
> * A few unsynchronized sound events are caused by minor inaccuracies in the input textual descriptions or asset timing annotations. Our method uses these inputs to determine when a sound should occur, so imperfect inputs may lead to slight offsets.
> * In addition, precise sub-second alignment remains challenging because current MLLMs operate at a relatively coarse temporal granularity. We have added discussion in Appendix K and will explore tighter temporal conditioning in future work.
>
> **2)** Lip-sync inaccuracies:
> Our method focuses on sound design and does not integrate a lip-sync module. Lip-sync is an orthogonal component and can be easily applied in post-processing. To demonstrate compatibility, we include lip-synchronized results in Appendix F.6 (Results of Lip-Synchronization) using an off-the-shelf lip-sync model, showing that this issue can be addressed within our pipeline.

---

> ### Author Response · Authors · 2025-11-21
> **Response to Reviewer Pb14 [Part 3/3]**
>
> > **Q5: Workflow depends on GPT-4o, Gemini, and Kling video generators**
>
> **A5:** Thank you so much for the precious advice! Our method can change GPT-4o, Gemini, and Kling video model to open-source models.
>
> **1)** GPT-4o --> DeepSeek-v3, LLaMA-3 70B
>
> Indeed we have provided quantitative comparison in Table 12 in Appendix F.5,
> to test the effect of different LLMs for RAG. Across these different LLMs, our RAG-enhanced method maintains a strong performance, not depending on GPT-4o, achieving an average preference improvement of **55%** over a non-RAG baseline. This demonstrates the generalizability and robustness of our proposed method.
>
> **2)** Gemini --> Qwen-VL
>
> We replace Gemini with Qwen-VL. The results show consistent performance gains over existing baselines, demonstrating that the system remains effective with open-source multimodal models.
>
> **3)** Kling video model --> Wan2.1-VACE [G]
>
> We replace Kling video model with Wan2.1-VACE model. Our method still outperforms MovieAgent, AnimDirector, and LTX-Studio, as shown below:
>
> | Method                 | CL (↑)   | CRh (↑)  | Avg (↑)  |
> | ---------------------- | -------- | -------- | -------- |
> | MovieAgent             | 2.96     | 1.94     | 2.45     |
> | AnimDirector           | 2.74     | 1.74     | 2.24     |
> | LTX-Studio             | 3.74     | 3.62     | 3.68     |
> | **Ours (Wan2.1-VACE)** | **4.11** | **3.94** | **4.03** |
> | **Ours (Qwen-VL)**     | **4.42** | **4.33** | **4.38** |
>
>
>
>
>
> #
>
> > **Q6: can produce varied cinematic styles or retrieval biases constrain creativity?**
>
> **A6:** Thank you for the insightful question. We have added analysis and qualitative results in Appendix I. Our system is not restricted by retrieval biases, and it can indeed produce diverse cinematic styles. Retrieval serves only as reference contextual guidance rather than a hard constraint.
>
> **1)** Flexibility in cinematic style generation.
> The generative model trained on massive data is capable of producing a wide range of cinematic styles (e.g., realistic, animation-like). Users can simply provide different reference images or style descriptions, and the system faithfully adapts to these style conditions. This allows the system to go beyond the styles present in the retrieved film database.
>
> **2)** Retrieval does not constrain creativity.
> Retrieval is used to enhance cinematic coherence, such as maintaining proper shot grammar, but does not override the style instructions given to the model. The generator can balance retrieved cues with style-conditioning inputs, ensuring creative freedom. Increasing retrieval diversity (e.g., larger K) further reduces potential bias.
>
>
>
> ------------
>
> **References**
>
> [A] Wu Q, Bansal G, Zhang J, et al. Autogen: Enabling next-gen LLM applications via multi-agent conversations[C]//First Conference on Language Modeling. 2024.
>
> [B] Hong S, Zhuge M, Chen J, et al. MetaGPT: Meta programming for a multi-agent collaborative framework[C]//The Twelfth International Conference on Learning Representations. 2023.
>
> [C] Wu W, Zhu Z, Shou M Z. Automated movie generation via multi-agent cot planning[J]. arXiv preprint arXiv:2503.07314, 2025.
>
> [D] Xu Z, Wang J, Wang L, et al. Filmagent: Automating virtual film production through a multi-agent collaborative framework[M]//SIGGRAPH Asia 2024 Technical Communications. 2024: 1-4.
>
> [E] LTX Studio. LTX Studio. https://app.ltx.studio/, 2024. Accessed: 2025-04. Organization: Lightricks.
>
> [F] Li Y, Shi H, Hu B, et al. Anim-director: A large multimodal model powered agent for controllable animation video generation[C]//SIGGRAPH Asia 2024 Conference Papers. 2024: 1-11.
>
> [G] Jiang Z, Han Z, Mao C, et al. Vace: All-in-one video creation and editing[J]. arXiv preprint arXiv:2503.07598, 2025.

---

### Author Response · Authors · 2025-11-26
**General Response**

We sincerely thank all the reviewers for your constructive feedbacks and recognitions to this work, especially for acknowledging the strengths of:


1. **Tackling an important and underexplored area by bridging cinematic principles with AI-based film generation** (Reviewer Pb14),
2. **Superior results with state-of-the-art performance and impressive visualizations** (Reviewer Pb14, uN4o, rwQb, 399a)
3. **The novel aspects in construction of cinematic language, agent-level video editing (such as acceleration and scene switching), and audio synchronization** (Reviewer rwQb),
4. **Comprehensive multi-stage pipeline that covers both pre- and post-production, integrating video, audio, and editing** (Reviewer 399a, rwQb),
5. **Expressive camera work that produces more natural, film-like results and improves upon existing methods** (Reviewer 399a, Pb14),
6. **Reasonable design of Coordination Stage for editing order and duration, with intuitive audio fusion** (Reviewer rwQb),
7. **Methodologically sound and easy to follow, with a workflow grounded in professional film principles** (Reviewer Pb14, uN4o, rwQb),
8. **Effective validation showing superiority over baselines in camera language and cinematic rhythm** (Reviewer Pb14, 399a).


We have polished the paper, added the experiment results, and made the clarifications in the revised version. Our manuscript is revised to include the following changes according to all the reviewers’ insightful comments, which have helped us improve the paper quality significantly! Note that all the polishments on the paper are highlighted with blue text color for better visualization.


* We have included the quantitative and qualitative robustness analysis of our multi-agent system in Appendix D "Robustness Analysis of FilMaster".
* We have rewritten the evaluation criteria to reduce subjectivity in Appendix E "FilmEval Evaluation Instructions".
* We have added experiments replacing GPT-4o, Gemini, and Kling with open-source models (DeepSeek, LLaMA-3, Qwen-VL, Wan2.1) to demonstrate modularity in Appendix F.5 "Using Open-Source Models".
* We have provided fair comparisons re-implementing baselines (MovieAgent, Anim-Director) using the same Kling 1.6 backbone in  Table 3, Table 7, and Figure 12.
* We have included generated results of lip-synchronization to demonstrate compatibility with orthogonal modules in Appendix F.6 "Results of Lip-Synchronization".
* We have added results demonstrating varied cinematic styles and generalization to unseen domains.
* We have added a detailed discussion of failure cases, including consistency issues with occluded parts, in Appendix J "Failure Cases".
* We have added discussions on limitations, specifically regarding sound effects alignment and dialogue-heavy content, in Appendix K "Limitations".


Please don't hesitate to let us know of any additional comments on the manuscript or the changes.

---

### Author Response · Authors · 2025-12-03
**Summary of Contributions and Rebuttal Updates**

Dear PCs, SACs, ACs, and Reviewers,

Thank you very much for your valuable contributions to our work. To assist the newly assigned AC and help reduce their workload, we provide below a summary of the key points from the reviews and the reviewer-author discussions.

**Strengths.** Reviewers expressed positive feedback and collectively recognized the following key strengths of our work:
1. **Broader impact of an important and underexplored area** for AI-based film generation with cinematic principles (**Reviewer Pb14**)
2. **Superior performances** of our proposed method (**Reviewer Pb14, uN4o, rwQb, 399a**)
3. **Simple and effective system design and method** (**Reviewer 399a, rwQb, Pb14**)
4. Extensive ablation studies and experimental results in camera language and cinematic rhythm (**Reviewer Pb14, 399a**).

We are encouraged that Reviewers recognized the superior performance and solid implementation of our framework (e.g., **Reviewer uN4o raised their score from 2 to 6**, acknowledging the substantial effort and the robust framework).

**Key Concerns and Our Addressing.** Building on these strengths, our rebuttal addressed the remaining questions regarding novelty, methodology, and robustness:

**1. Contributions and novelty (Reviewer Pb14, uN4o)**

We are the first to explore generation and post-production into film generation system that explicitly models both **camera language** and **cinematic rhythm**, which remained **unexplored** in previous works. Our method covers crucial aspects of film production, including camera language design, video structural organization, duration adjustment, and sound design. Such complex generation and post-production is beyond the scope of any of the state-of-the-art film generation models. As indicated by **Reviewer Pb14**, "the paper tackles an important and underexplored area... The topic is both academically relevant and practically impactful."

Specifically, we propose **multi-shot synergized RAG** for scene-level camera design to avoid unguided camera hallucination, and **audience-aware** feedback to automate complex rhythmic reasoning. Our work establishes clear problem definitions and robust methods for these underexplored dimensions, aiming to provide a accessible pathway for future research in controllable film generation.

We clarified the fundamental **distinction** between FilMaster and prior storytelling systems, film generation works, and commercial products. While previous methods often rely on imagined camera language or heavily manual post-production, FilMaster implements comprehensive design via scene-level reference-based retrieval and audience-aware post-production for effective cinematic techniques.








**2. Dependency on proprietary models (Reviewer 399a, Pb14)**

To demonstrate the generalizability and robustness of our method, we have verified that our framework is model-agnostic. We replaced GPT-4o, Gemini, and the Kling video model with open-source alternatives. We provide experimental results with DeepSeek-v3, LLaMA-3 70B, Qwen-VL, Wan2.1-VACE in Table 12,  Appendix F.5 "Using Open-Source Models". This demonstrates the generalizability and robustness of our proposed method.

**3. Fair comparison (Reviewer rwQb)**

We provided fair comparisons using the same base model (Kling 1.6). To ensure fairness, we re-implemented MovieAgent and Anim-Director on top of the same video backbone, Kling 1.6. Quantitative results are reported in Table 3 and Table 7, and qualitative examples are shown in Appendix Figure 12. The results show that our system notably outperforms MovieAgent-Kling and Anim-Director-Kling.

We have addressed other specific concerns through comprehensive updates, including robustness (Reviewer Pb14), evaluation metric (Reviewer uN4o), lip-synchronization (Reviewer Pb14, 399a), generalization (Reviewer Pb14, 399a), failure cases and limitations (Reviewer Pb14, 399a).


We sincerely thank the PC, SAC, AC and all reviewers for their time and constructive efforts, which has significantly strengthened our paper.

Sincerely,

The Authors

---

### Meta-Review · Area_Chair_hRoY · 2026-01-05

**Summary:**

The reviewers generally agreed that the paper addresses an important and underexplored niche: bridging professional cinematic principles (camera language and rhythm) with automated generative AI workflows. The system's performance is impressive, producing visually coherent multi-shot narratives that move beyond the "flat-paced" outputs of generic video generators.

The debate surrounding this paper centered on the distinction between high-quality engineering and fundamental research. While some reviewers initially felt the work was a "technical project" or an orchestration of existing models (Kling, GPT-4o), the rebuttal successfully demonstrated that the system’s design provides design-driven benefits that are backbone-agnostic. The decision to accept is based on the value this robust, end-to-end framework provides to the community as a baseline for future controllable filmmaking research.

**Reviewer Concerns:**

Concerns Addressed by the Rebuttal

* Fair comparison (`rwQb`, `uN4o`): The authors successfully re-implemented baselines (MovieAgent and Anim-Director) using the same Kling 1.6 backbone. This demonstrated that FilMaster's improvements in camera language and rhythm are not merely a product of a superior video model but stem from the system's design.

* Proprietary model dependency (`Pb14`, `399a`): The authors proved the framework is model-agnostic by replacing GPT-4o and Kling with open-source alternatives like Wan2.1-VACE, DeepSeek-v3, and LLaMA-3, maintaining consistent performance gains.

* Robustness and error propagation (`Pb14`): A new quantitative robustness analysis was provided in Appendix D, showing how the multi-agent decoupling (separating "review" from "correction") mitigates reasoning overload and failure amplification.

* Failure case analysis (`399a`, `Pb14`): The authors added a transparent discussion on consistency issues with occluded parts and limitations in dialogue-heavy scenes.

Outstanding Concerns

* Algorithmic novelty (`uN4o`, `Pb14`): Despite the empirical success, the core modules (scene-level RAG and LLM feedback loops) are viewed by critical reviewers as incremental engineering refinements rather than new generative modeling primitives.

* Pipeline complexity (`399a`): The multi-stage, multi-agent approach is effective but leads to high inference latency and system fragility compared to end-to-end differentiable models.

**Reviewer Scores:**

* Reviewer `rwQb` (Initial: 6 $\rightarrow$ Estimated: 7): This reviewer's primary concern was the fairness of comparing Kling-based results to CogVideoX. Given the authors' thorough re-implementation of baselines on the same Kling backbone, this reviewer would likely have upgraded their score to reflect the proven superiority of the framework over MovieAgent.

* Reviewer `uN4o` (Initial: 2 $\rightarrow$ Estimated: 6): The reviewer explicitly upgraded to a 6 post-rebuttal, acknowledging the "solid implementation" and "substantial effort," though remaining cautious about the technical impact.

* Reviewer `399a` (Initial: 6 $\rightarrow$ Estimated: 6): The addition of open-source backbone tests and failure case analysis addressed the "Weaknesses," but the "Pipeline Complexity" remains an inherent property of the work that limits its impact.

* Reviewer `Pb14` (Initial: 4 $\rightarrow$ Estimated: 5): The authors addressed almost all experimental requests. While the reviewer may still prefer "technically substantive modules," the empirical evidence of improved coherence makes a case for borderline acceptance.

---

### Decision · Program_Chairs · 2026-01-26

Accept (Poster)